# Multisensory input modulates memory-guided spatial navigation in humans

Deetje Iggena [1,2,7✉], Sein Jeung[3,4,5,7], Patrizia M. Maier[1,2], Christoph J. Ploner[1], Klaus Gramann [3,6] & Carsten Finke [1,2]

Efficient navigation is supported by a cognitive map of space. The hippocampus plays a key role for this map by linking multimodal sensory information with spatial memory representations. However, in human navigation studies, the full range of sensory information is often unavailable due to the stationarity of experimental setups. We investigated the contribution of multisensory information to memory-guided spatial navigation by presenting a virtual version of the Morris water maze on a screen and in an immersive mobile virtual reality setup. Patients with hippocampal lesions and matched controls navigated to memorized object locations in relation to surrounding landmarks. Our results show that availability of multisensory input improves memory-guided spatial navigation in both groups. It has distinct effects on navigational behaviour, with greater improvement in spatial memory performance in patients. We conclude that congruent multisensory information shifts computations to extrahippocampal areas that support spatial navigation and compensates for spatial navigation deficits.

[1] Charité - Universitätsmedizin Berlin, Department of Neurology, Augustenburger Platz 1, 13353 Berlin, Germany. [2] Humboldt-Universität zu Berlin, Berlin School of Mind and Brain, Unter den Linden 6, 10099 Berlin, Germany. [3] Technische Universität Berlin, Department of Biological Psychology and Neuroergonomics, Fasanenstraße 1, 10623 Berlin, Germany. [4] Norwegian University of Science and Technology, Kavli Institute for Systems Neuroscience, Olav Kyrres gate 9,7030, Trondheim, Norway. [5] Max-Planck Institute for Human Cognitive and Brain Sciences, Stephanstraße 1a, 04103 Leipzig, Germany. [6] University of California, San Diego, Center for Advanced Neurological Engineering, 9500 Gilman Dr, La Jolla, CA 92093, USA. [7] These authors contributed equally: Deetje Iggena, Sein Jeung. ✉email: deetje.iggena@charite.de

The ability to navigate distinct environments and locations from memory is a prerequisite for autonomy and survival. For effective navigation, we continuously update our position and orientation by integrating multisensory information with memory representations of the environment[1–3]. Relevant sensory information includes visual, vestibular, and proprioceptive input[4–8]. These inputs are integrated and transformed into spatial representations, which are encoded, consolidated, and eventually recalled, in formats that depend on actual behavioral demands[9,10].

One of the core regions in brain networks for spatial navigation is the hippocampus. The hippocampus binds and integrates location-specific information from multiple sensory modalities and uses it to transform spatial relationships into a global cognitive map[11,12]. During the formation process of this map, the hippocampus is in constant exchange with other brain regions and shares computations with the parahippocampal, entorhinal, and the retrosplenial cortex, among others[1,13]. Accordingly, behavioral assessment of spatial navigation has become an important tool to test hippocampal function across species. In animal models, particularly in rodents, navigation in environments such as the Morris Water Maze (MWM) is a widely used standard in spatial memory research[14,15]. However, in many human navigation experiments, memory-guided spatial navigation is mainly investigated with moving visual stimuli on a stationary screen, without the vestibular, and proprioceptive inputs that contribute to behavior in animal experiments. The absence of these body-based sensory inputs creates an artificial situation that limits spatial information for navigation and may promote behaviors that do not necessarily reflect everyday demands in human participants[16,17]. In particular, for humans with structural or functional lesions in brain regions critical for spatial navigation, alternative behaviors may be triggered depending on the availability of body-based sensory cues[18–21]. The ecological validity of stationary navigation paradigms has therefore been repeatedly questioned[22–24].

Advances in mobile immersive virtual reality (VR) technologies provide an opportunity to overcome these limitations and to study human spatial navigation with true multisensory input[25–27]. Head-mounted displays (HMDs) immerse participants in a highly realistic yet controlled virtual environment, in which they can move freely, generating and receiving ample body-based sensory input. Mobile VR systems, therefore enable the test conditions for humans that are largely analogous to animal experiments. This allows a more direct comparison of data from humans with data from freely moving rodents obtained in navigation tasks such as the MWM.

The present study aimed to systematically investigate the contribution of multisensory information to human spatial navigation. We asked whether and how memory-guided spatial navigation benefits from multisensory input in humans with and without hippocampal dysfunction. Particularly, we were interested in how hippocampal lesions alter the use of multisensory information and whether this is reflected in altered navigational behavior. To this end, we implemented a virtual MWM task and compared the navigation patterns of patients with hippocampal lesions to those of healthy controls in both a stationary desktop and in a room-scale mobile VR environment.

We found that the availability of multisensory information improved memory-guided spatial navigation in both patients and healthy controls, with greater improvement in spatial memory performance in patients. This improved memory performance was accompanied by different navigation behavior in patients and healthy controls. This suggests that multisensory information may shift computations to extrahippocampal areas that support spatial navigation and compensate for hippocampus-related deficits in spatial navigation.

## Results

To evaluate the effects of multisensory input on memory-guided spatial navigation, we examined spatial memory performance and navigation behavior in a virtual version of the MWM. While a water maze, is typically implemented in rodent studies in a pool filled with water, we built a dry version of human scale water maze, using a virtual circular enclosure filled with virtual ground fog. It was presented either on a screen where participants navigated the environment with a joystick (stationary), or in a room-scale immersive VR setup where participants could walk freely during the task, (mobile), (Fig. 1a, b, see methods). The session order of experimental setups was counterbalanced to account for potential learning effects from the first experimental setup that could influence navigation behavior in the second experimental setup. The circular arena of the water maze was surrounded by landmarks embedded in a natural-looking hilly landscape. We counterbalanced the design of the environment between the two experimental setups stationary and mobile (Fig. 1c).

Patients with hippocampal lesions (MTLR, $n = 10$) and their healthy controls (Control, $n = 20$) learned the locations of objects by exploring the water maze, repeatedly starting from the same location. During three learning trials, the object appeared as soon as the target location was reached (Fig. 1d). After the three learning trials, four probe trials followed from four different starting locations to encourage the use of spatial relations in the environment. In probe trials, the object remained hidden, and participants indicated where they remembered the hidden object by pressing a button. Per experimental setup, six target objects were placed at different distances to the boundary to discourage the use of a circling strategy around the arena and in a distinct angular relation to the landmarks, to promote triangulation between landmarks and the target locations (Supplementary Table 1)

After each learning or probe trial, a disorientation task followed that forced self-localization at the onset of each trial in both stationary and mobile setups. Briefly, in this task, all spatial cues were blanked out. Participants first navigated to spheres that triggered a random sequence of three turns, then they were led by spheres to the starting point of the next trial and the spatial features of the virtual environment reappeared (see methods and Supplementary methods 1). By computing how close the final response location was to the target location, we assessed spatial memory performance, and by inferring behavioral patterns from the path traveled, we analyzed navigation efficiency and navigation strategies (see methods).

**Multisensory input improves spatial memory performance in patients with hippocampal lesions**. We investigated how accurately participants retrieved the learned object locations during probe trials. This aspect of spatial navigation depends on the integrity of spatial memory representations and the ability to determine one's location relative to the environment.

We calculated the distance between the final location of the participants and the actual target location to derive a memory score, taking into account the geometry of the environmental boundary[28,29], (Fig. 2a). The memory score ranges from 0 to 100%, with values close to 100% suggesting that the participant could perfectly remember and locate the target location, 50% suggesting that the final location was chosen at chance level, and close to 0% that the participant's final position was systematically biased in the opposite direction of the actual target location (see methods).

Both patients with medial temporal lobe resections including the hippocampus and healthy controls, performed above chance level in both the stationary desktop and the mobile VR setup (Fig. 2b). In the stationary setup, patients had a lower average memory score compared with their healthy controls

**a** Stationary setup

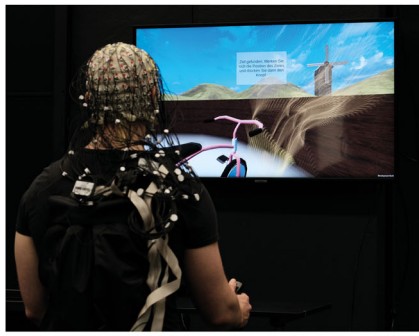

**b** Mobile setup

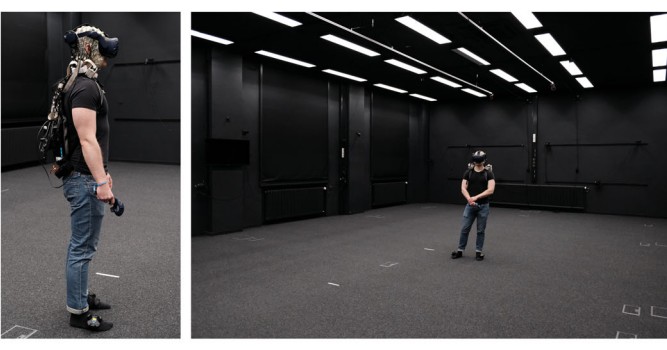

**c** Panorama view scene A

Panorama view scene B

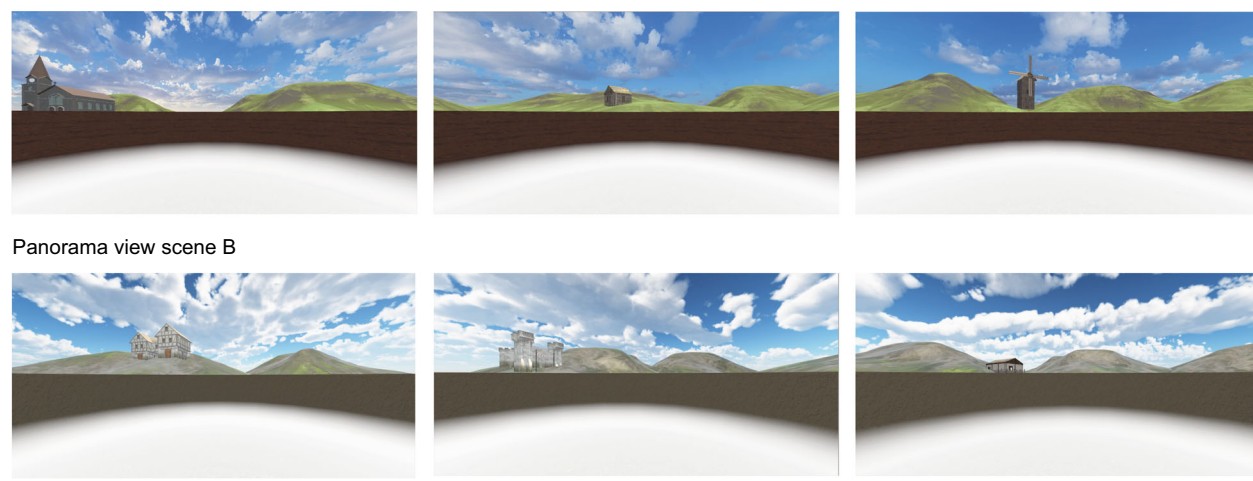

**d**

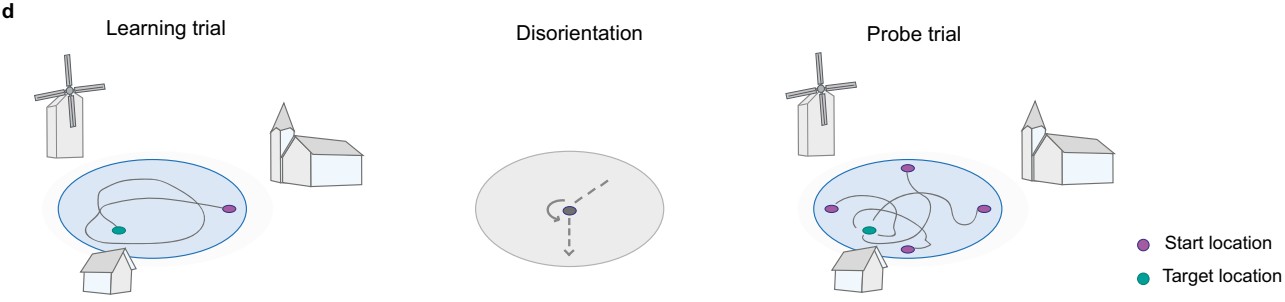

**Fig. 1 Experimental setup. a** Exemplary images of stationary setup where the participant stands in front of a screen. **b** Exemplary images of mobile setup where the participant wears head-mounted display glasses to experience VR. **c** Exemplary participant view of the virtual environments. Two different scenes were created for counterbalanced design across the two experimental setups stationary and mobile. **d** Experimental block design. Each block started with three learning trials in which the starting location remained constant, and the object appeared as feedback for participants. Learning trials were followed by four probe trials, in which the starting locations varied with rotations around the center by 0, 90, 180, or 270°, and participants had to indicate the remembered object location. A total of six object locations were learned in each setup. All consecutive trial pairs were separated by a spatial disorientation task.

(Mean ± SEM: 72.4 ± 5.4 vs. 87.6 ± 2.1; Supplementary Table 2). However, when participants had access to multisensory information in the mobile setup, the memory score increased by 23.6% in patients (Mean ± SEM: 72.4 ± 5.4 to 89.5 ± 1.8) and by 7.6% in controls (Mean ± SEM: 87.6 ± 2.1 vs. 94.6 ± 1.0). Although memory performance of both groups benefited from multisensory information, the change in memory score across setups was significantly more pronounced in patients compared to healthy controls (setup*group: $F_{(1,27)} = 5.207$, $p = 0.031$, $\omega^2 = 0.13$; stat-MTLR vs. mobile-MTLR, $p < 0.001$; stat-control vs. mobile-

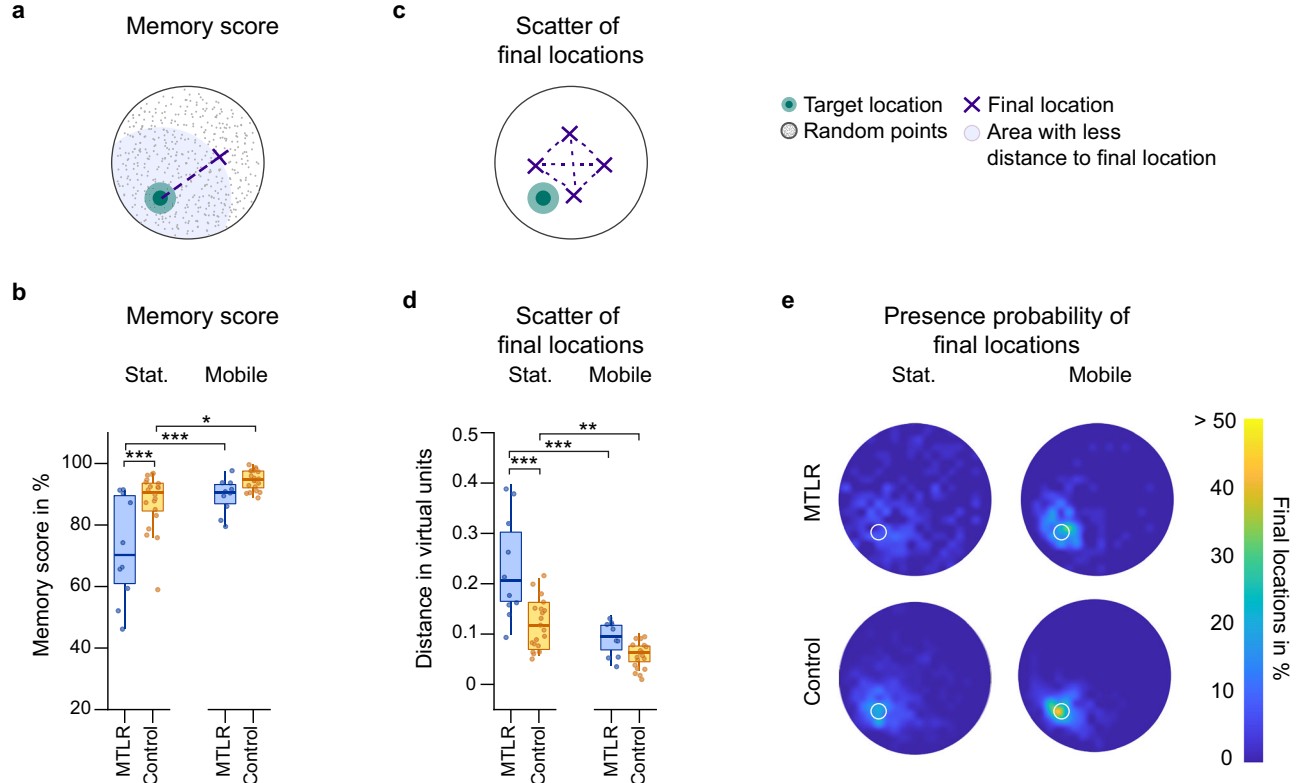

**Fig. 2 Spatial memory performance and spatial precision in probe trials. a** Schematic representation of the calculation of the memory score. 1000 random locations with uniform spatial distribution were generated. The percentage of locations with smaller distance to the target location than the participant's final chosen location was subtracted from 100%. 50% corresponds to random-level performance, a higher memory score indicates a bias towards the target location and a score below 50% a bias towards the opposite direction of the target location. **b** The memory score as a measure of spatial memory performance. Patients benefited more from access to multisensory information and showed a greater increase in memory performance than the control group (setup*group: $F_{(1,27)} = 5.207$, $p = 0.031$, $\omega^2 = 0.13$). **c** Schematic representation of the calculation of the scatter of the final locations. The distance between each final location pair was averaged across all distances per target location. A value closer to zero indicates less scatter of the final locations and thus higher spatial precision. **d** Scatter of final locations as a measure of spatial precision. Patients benefited more from access to multisensory information and showed a greater decrease in scatter of final locations than the control group (setup*group: $F_{(1,52)} = 9.595$, $p = 0.003$, $\omega^2 = 0.14$) **e** Final locations in the arena are shown as a percentage at each location. Yellow indicates that more than half of the responses occurred at that location; dark blue indicates that no responses occurred at that location. Target location is marked with a white circle. Metric data presented as boxplots with a center line as median, Tukey-style whiskers extend 1.5 times the interquartile range from the 25th and 75th percentiles. Dots present individual datapoints. Data was analyzed with a linear mixed model. Sample size, medial temporal lobe resection (MTLR) group: $n = 10$, control: $n = 20$; * $= p \leq 0.05$; ** $= p \leq 0.01$; *** $= p \leq 0.001$.

control, $p = 0.031$; stat-MTLR vs. stat-control, $p < 0.001$; mobile-MTLR vs. mobile-control, $p = 0.251$).

**Multisensory input improves spatial precision in patients with hippocampal lesions**. We then investigated the precision of the representations underlying memory-guided navigation, a property that also depends on hippocampal integrity[30,31]. Regardless of the distance to the target location, precision indicates how consistent responses were per target location. As a measure of precision, we computed the scatter of participants' responses by calculating the relative distance of all six distances between the four final locations per target location in the probe trials. A smaller scatter meant higher spatial precision (Fig. 2c–e, see methods).

In the stationary setup, patients showed a higher average scatter in final locations compared with their healthy controls (Mean ± SEM: 0.23 ± 0.03 vs. 0.12 ± 0.01; Supplementary Table 2). When multisensory input was available in the mobile setup, the scatter decreased in both groups, by 59.9% in patients (Mean ± SEM: 0.23 ± 0.03 vs. 0.09 ± 0.01) and by 48.7% in controls (Mean ± SEM: 0.12 ± 0.01 vs. 0.06 ± 0.00). As with the memory score, the change in spatial precision was significantly more pronounced in patients

compared to healthy controls (setup*group: $F_{(1,52)} = 9.595$, $p = 0.003$, $\omega^2 = 0.14$; stat-MTLR vs. mobile-MTLR, $p < 0.001$; stat-control vs. mobile-control, $p = 0.002$; stat-MTLR vs. stat-control, $p < 0.001$; mobile-MTLR vs. mobile-control, $p = 0.194$).

**Multisensory input improves spatial navigation efficiency in patients with hippocampal lesions**. Analysis of navigation path to a location reveals various behavioral properties, such as the temporal and spatial efficiency of navigation. The better one can locate themselves and the target location in relation to landmarks, the faster and more directly the targeted destination can be reached.

The temporal efficiency of navigation can be assessed by the latency to the final location (Fig. 3a, b, see methods). We found that temporal efficiency increased in both, patients and healthy controls when multisensory input was available in the mobile VR setup. In learning trials, the improved temporal efficiency in the mobile setup compared to the stationary setup was reflected in a reduction in average latency to final location by 48.8% in patients (Mean ± SEM: 46.7 ± 17.4 vs. 23.9 ± 4.9, Supplementary Table 3) and by 22.2% in controls (Mean ± SEM: 19.8 ± 2.3 vs.

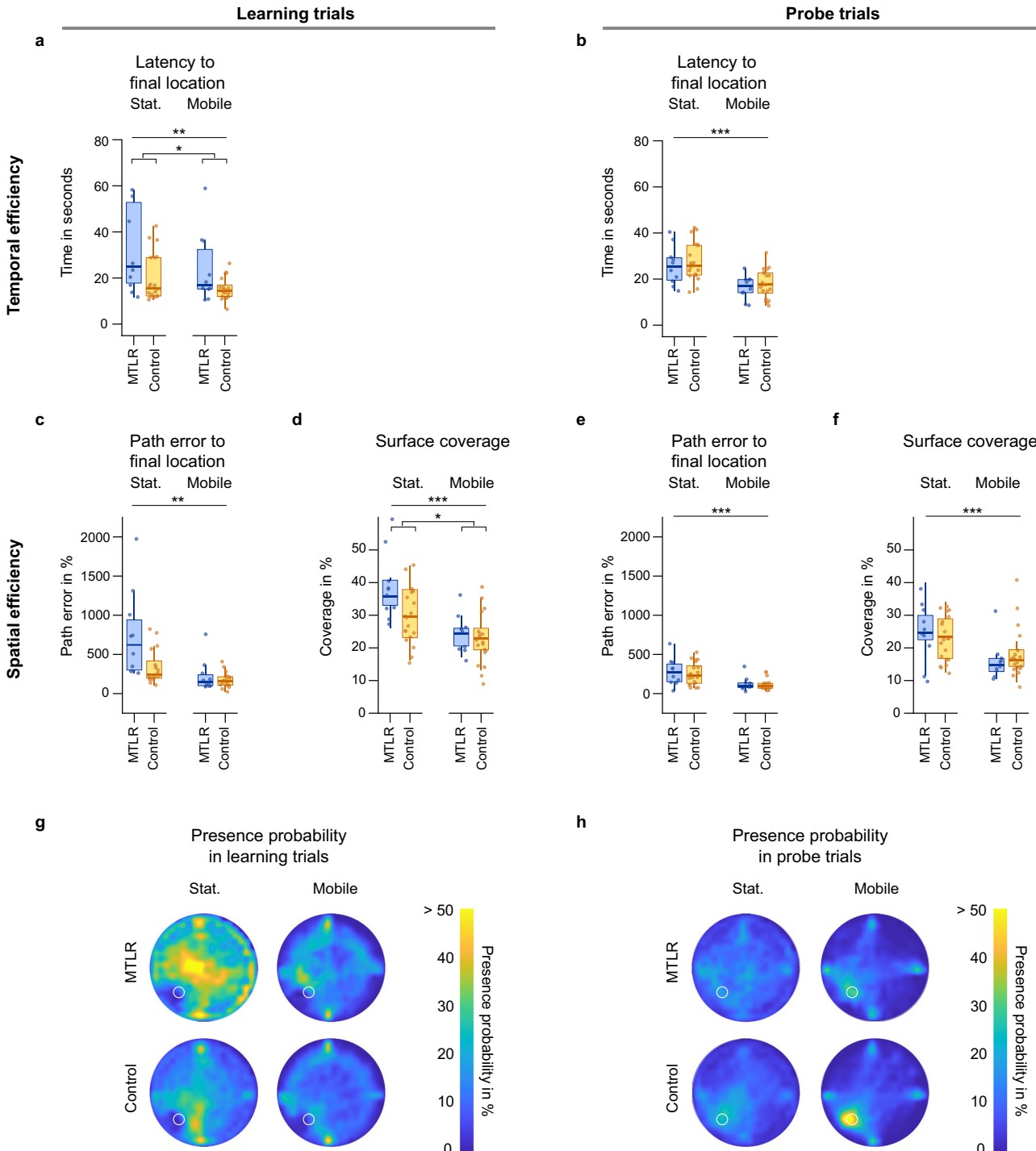

15.4 ± 1.1; setup: $F_{(1,27)} = 7.310$, $p = 0.012$, $\omega^2 = 0.18$). Across experimental setups, patients required more time to reach the final location than the healthy controls in the learning trials (group: $F_{(1,25)} = 6.457$, $p = 0.018$, $\omega^2 = 0.17$). In probe trials, the improved temporal efficiency in the mobile setup was reflected in a reduction in latency by 36.4% in patients (Mean ± SEM: 25.8 ± 2.7 vs. 16.4 ± 1.6) and by 34.5% in controls (Mean ± SEM: 27.5 ± 1.8 vs. 18.0 ± 1.4; setup: $F_{(1,27)} = 52.153$, $p < 0.001$, $\omega^2 = 0.64$). Across experimental setups, patients had a similar latency to reach the final locations as controls in probe trials (group: $F_{(1,25)} = 0.664$, $p = 0.423$, $\omega^2 = 0.0$).

Spatial efficiency is reflected in the path error and surface coverage. The path error is calculated as the percentage of

deviation of the actual path from an ideal path to the final location, and surface coverage is determined by the percentage of the arena area covered during navigation (Fig. 3c–f, see methods). As with temporal efficiency, we found that spatial efficiency increased in both, patients and healthy controls when multi-sensory input was available in the mobile VR setup (Fig. 3g, h). In learning trials, the improved spatial efficiency in the mobile setup was reflected in a decrease in average path error by 69.7% in patients (Mean ± SEM: 745.5 ± 175.5 vs. 226.1 ± 65.1) and by 67.9% in controls (Mean ± SEM: 530.6 ± 207.3 vs. 170.5 ± 21.0; setup: $F_{(1,52)} = 7.900$, $p = 0.007$, $\omega^2 = 0.11$). Across experimental setups, patients had a similar path error as controls in learning trials (group: $F_{(1,52)} = 0.630$, $p = 0.431$, $\omega^2 = 0.0$). In probe

**Fig. 3 Temporal and spatial navigation efficiency in learning and probe trials. a** Latency to final location as measure of temporal efficiency in learning trials. Multisensory input in the mobile setup increased temporal efficiency for both groups, as evidenced by reduced latency to final location (setup: $F_{(1,27)} = 7.310$, $p = 0.012$, $\omega^2 = 0.18$). **b** Latency to final location as measure of temporal efficiency in probe trials. Multisensory input in the mobile setup increased temporal efficiency for both groups, as evidenced by reduced latency to final location (setup: $F_{(1,27)} = 52.153$, $p < 0.001$, $\omega^2 = 0.64$). **c** Path error to final location as measure of spatial efficiency in learning trials. Multisensory input in the mobile setup increased spatial efficiency for both groups, as evidenced by reduced path error (setup: $F_{(1,52)} = 7.897$, $p = 0.007$, $\omega^2 = 0.11$). **d** Surface coverage as a measure of spatial efficiency in learning trials. Multisensory input in the mobile setup increased spatial efficiency for both groups, as evidenced by reduced path error (setup: $F_{(1,27)} = 33.499$, $p < 0.001$, $\omega^2 = 0.53$). **e** Path error to final location as measure of spatial efficiency in probe trials. Multisensory input in the mobile setup increased spatial efficiency for both groups, as evidenced by reduced path error (setup: $F_{(1,27)} = 48.153$, $p < 0.001$, $\omega^2 = 0.62$). **f** Surface coverage as measure of spatial efficiency in probe trials. Multisensory input in the mobile setup increased spatial efficiency for both groups, as evidenced by reduced path error (setup: $F_{(1,27)} = 26.654$, $p < 0.001$, $\omega^2 = 0.47$). **g, h** Presence probability in learning trials and probe trials. Presence probability describes the probability of being in a cell of a 20*20 grid covering the arena surface. Yellow means that the paths passed through that location in more than half of the trials, dark blue means that the paths did not pass the underlying cell of the grid at all. Target location is marked with a white circle. Metric data presented as boxplots with a center line as median, Tukey-style whiskers extend 1.5 times the interquartile range from 25th and 75th percentiles. Dots present individual datapoints. Data was analyzed with a linear mixed model. Sample size, MTRL: $n = 10$, control: $n = 20$; $* = p \leq 0.05$; $** = p \leq 0.01$; $*** = p \leq 0.001$.

trials, we found a decrease in path error by 53.6% in patients (Mean ± SEM: 278.8 ± 55.1 vs. 124.3 ± 28.1) and by 53.7% in controls (Mean ± SEM: 251.2 ± 31.9 vs. 116.2 ± 15.5; setup: $F_{(1,27)} = 48.153$, $p < 0.001$, $\omega^2 = 0.62$) in the mobile setup. Across experimental setups, patients had a similar path error as controls in probe trials (group: $F_{(1,25)} = 0.265$, $p = 0.611$, $\omega^2 = 0.0$).

The improved spatial efficiency was also reflected in lower surface coverage when multisensory input was available in the mobile VR setup. In learning trials, average surface coverage decreased by 36.9% in patients (Mean ± SEM: 38.8 ± 3.5 vs. 24.5 ± 1.8) and by 23.5% in controls (Mean ± SEM: 30.2 ± 1.9 vs. 23.1 ± 1.6; setup: $F_{(1,28)} = 33.499$, $p < 0.001$, $\omega^2 = 0.53$). Across experimental setups, patients had a similar surface coverage as controls in learning trials (group: $F_{(1,25)} = 4.755$, $p = 0.039$, $\omega^2 = 0.12$). In probe trials, we found a decrease in surface coverage by 34.9% in patients (Mean ± SEM: 24.9 ± 2.8 vs. 16.2 ± 1.9) and by 18.8% in controls (Mean ± SEM: 22.9 ± 1.6 vs. 18.6 ± 1.7; setup: $F_{(1,27)} = 26.254$, $p < 0.001$, $\omega^2 = 0.47$) in the mobile setup. Across experimental setups, patients had a similar surface coverage as controls in probe trials (group: $F_{(1,25)} = 0.008$, $p = 0.929$, $\omega^2 = 0.0$).

In contrast to spatial memory performance and navigation strategies, we found an influence of the session order on navigation efficiency, at least for the performance in probe trials. The result indicates that with increasing experience with the task itself navigation efficiency increases (Probe trials: latency, $F_{(1,27)} = 8.096$, $p = 0.008$, $\omega^2 = 0.20$; path error, $F_{(1,27)} = 21.206$, $p < 0.001$, $\omega^2 = 0.41$; surface coverage, $F_{(1,27)} = 12.404$, $p = 0.002$, $\omega^2 = 0.28$; see Supplementary Table 4).

**Multisensory input modulates navigation strategies in patients with hippocampal lesions**. To achieve the goal of navigating to the target location, a range of different strategies were available to participants. We used the observed movement patterns, such as the shape of the path to a location as well as the rotational behavior of the navigators, to infer on the underlying navigation strategies. We extracted three parameters that reflect different strategies that participants employed to find the target in the water maze: search accuracy, landmark use, and path replication. The choice of one strategy does not preclude the use of other strategies, as participants may switch between strategies and use more than one strategy simultaneously on the way to the target location[21,32].

Search accuracy describes the spatial focus of the search behavior in the water maze. It is characterized by the average distance to the final location[33,34], (Fig. 4a–c). A lower average distance reflects a preference for more intensive and focused search of the object near the final location, while higher averaged distance is found when participants are primarily searching randomly or distant from the final location (see methods).

In learning trials, we found that the availability of multi-sensory input in the mobile VR setup led to greater improvement in search accuracy in patients than in controls. The increase was reflected in a decrease in the average distance to the final location by 19.0% in patients (Mean ± SEM: 0.42 ± 0.01 vs. 0.34 ± 0.01) and by 7.9% in controls (Mean ± SEM: 0.38 ± 0.01 vs. 0.35 ± 0.01; group*setup: $F_{(1,52)} = 4.456$, $p = 0.040$, $\omega^2 = 0.06$; stat-MTLR vs. mobile-MTLR, $p = 0.003$; stat-control vs. mobile-control, $p = 0.163$; stat-MTLR vs. stat-control, $p = 0.081$; mobile-MTLR vs. mobile-control, $p = 0.566$). In contrast, in probe trials, we found that the availability of multisensory input in the mobile VR setup increased search accuracy to a similar extent for both groups. The increase was reflected in a decrease in the average distance to the final location by 15.2% in patients (Mean ± SEM: 0.33 ± 0.02 vs. 0.28 ± 0.02) and by 16.7% in controls (Mean ± SEM: 0.30 ± 0.01 vs. 0.25 ± 0.02; setup: $F_{(1,27)} = 22.429$, $p < 0.001$, $\omega^2 = 0.42$). Across experimental setups, patients had a similar search accuracy as controls in probe trials (group: $F_{(1,25)} = 1.644$, $p = 0.212$, $\omega^2 = 0.02$).

The use of landmarks is the most efficient strategic behavior in an allocentric spatial navigation task such as the MWM[15,19]. Early incorporation of information from the surroundings, such as landmarks, accelerates self-localization, localization of the target location, and eventually the calculation of the optimal path. A measure of the estimated use of landmarks for path planning is the integrated absolute angular velocity at the start of a trial[35,36], (Fig. 4d–f). The measure describes the extent of head movements, and a higher value refers to more intense use of the surrounding landmarks (see methods).

In learning trials, we found that the availability of multisensory input in the mobile VR setup increased the use of landmarks to a similar extent in patients and controls. The increase in landmark use was reflected in an increase in average angular velocity by 31.1% in patients (Mean ± SEM: 0.0060 ± 0.0006 vs. 0.0081 ± 0.0005) and by 36.8% in controls (Mean ± SEM: 0.0076 ± 0.0004 vs. 0.0105 ± 0.0005; setup: $F_{(1,27)} = 32.830$, $p < 0.001$, $\omega^2 = 0.52$). Across experimental setups, patients had a lower angular velocity than controls in learning trials (group: $F_{(1,25)} = 9.206$, $p = 0.006$, $\omega^2 = 0.23$). In contrast, in probe trials, we found that the availability of multisensory input in the mobile VR setup increased the use of landmarks significantly less in patients than in controls. This was reflected in an increase in angular velocity by 122.2% in patients (Mean ± SEM: 0.0036 ± 0.0005 vs. 0.0080 ± 0.0004) and by 161.0% in controls (Mean ± SEM: 0.0041 ± 0.0003 vs. 0.0107 ± 0.0005; setup*group: $F_{(1,52)} = 5.375$, $p = 0.024$, $\omega^2 = 0.07$; stat-MTLR vs. mobile-MTLR, $p < 0.001$; stat-control vs. mobile-control, $p < 0.001$; stat-MTLR vs. stat-control, $p = 0.477$; mobile-MTLR vs. mobile-control, $p < 0.001$).

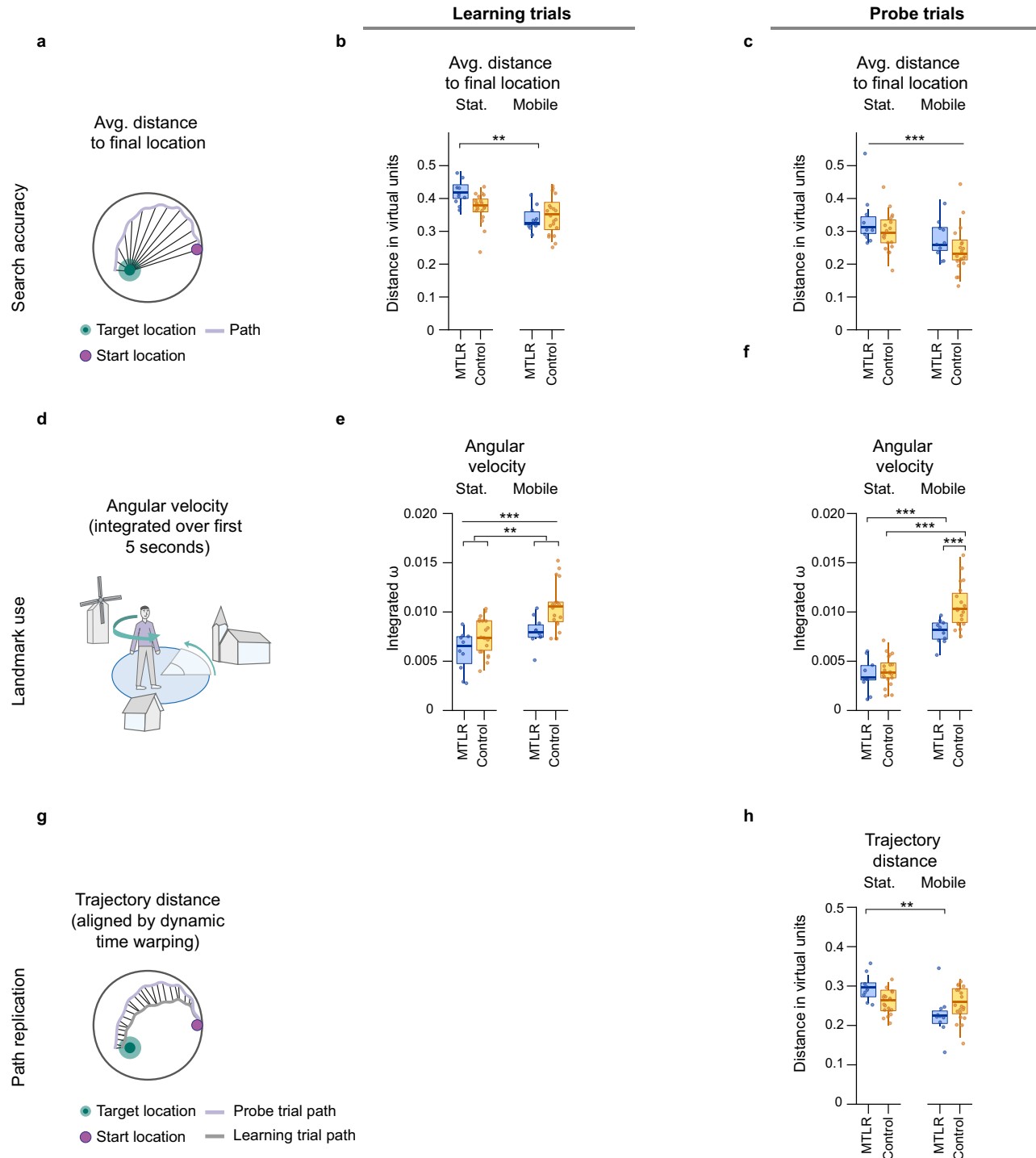

Path replication is a strategic behavior based on route-based learning. It is realized by repeatedly navigating to the location of an object from a fixed starting location in learning trials. The use of repeated path sequences to reach the final location relies on egocentric representations, even when approaching the target location from a new location[18,21,32,37]. The extent of repetition is reflected in the distance between the aligned trajectories of the last learning trial and the trajectories for each probe trial (Fig. 4g, h). A smaller distance between the trajectories reflects a stronger repetition of the path, while large distances represent dissimilar trajectories (see methods).

We found that the availability of multisensory input in the mobile VR setup led to replication of path sequences significantly more in patients than in controls. The increase in path replication was reflected by a decrease in trajectory distance by 23.3% in patients (Mean ± SEM: 0.30 ± 0.01 vs. 0.23 ± 0.02) and by 0.0% in controls (Mean ± SEM: 0.26 ± 0.01 vs. 0.26 ± 0.01; setup*group: $F_{(1,52)} = 10.723$, $p = 0.002$, $\omega^2 = 0.15$; stat-MTLR vs. stat-control, $p = 0.063$; mobile-MTLR vs. mobile-control, $p = 0.079$; stat-MTLR vs. mobile-MTLR, $p = 0.002$; stat-control vs. mobile-control, $p = 0.759$).

## Discussion

We investigated effects of multisensory input on memory-guided spatial navigation in humans with and without hippocampal

**Fig. 4 Navigation strategies in learning and probe trials. a** Schematic representation of the calculation of the average distance to the final location as a measure of search accuracy. **b** Multisensory input in the mobile setup increased patients' search accuracy more than controls search accuracy in learning trials (group*setup: $F_{(1,53)} = 4.442$, $p = 0.040$, $\omega^2 = 0.06$; stat-MTRL vs. mobile-MTLR, $p = 0.004$; stat-control vs. mobile-control, $p = 0.219$). **c** Multisensory input in the mobile setup increased patients' and controls' search accuracy to a similar extent in probe trials (setup: $F_{(1,28)} = 20.726$, $p < 0.001$, $\omega^2 = 0.40$). **d** Schematic representation of the calculation of the angular velocity over first five seconds as a measure for landmark use for self-localization, and path planning. **e** Multisensory input in the mobile setup increased patients' and controls' landmarks use to a similar extent in learning trials (setup: $F_{(1,27)} = 32.830$, $p < 0.001$, $\omega^2 = 0.52$), but patients used less landmarks across setups (group: $F_{(1,25)} = 9.206$, $p = 0.006$, $\omega^2 = 0.23$). **f** Multisensory input in the mobile setup increased patients' landmarks use less than controls resulting group differences in probe trials (setup*group: $F_{(1,53)} = 5.375$, $p = 0.024$, $\omega^2 = 0.07$; stat-MTLR vs. mobile-MTLR, $p < 0.001$; stat-control vs. mobile-control, $p < 0.001$; stat-MTLR vs. stat- control, $p = 0.477$; mobile-MTLR vs. mobile-control, $p < 0.001$). **g** Schematic representation of the calculation of the trajectory distance using dynamic time warping for alignment of trajectories to compare as a measure for path replication of the last learning trial trajectory and each probe trial trajectory. **h** Multisensory input in the mobile setup increased patients' use of path replication more than controls (setup*group: $F_{(1,52)} = 10.723$, $p = 0.002$, $\omega^2 = 0.15$; stat-MTLR vs. mobile-MTLR, $p = 0.002$; stat-control vs. mobile-control, $p = 0.759$; stat-MTLR vs. stat- control, $p = 0.063$; mobile-MTLR vs. mobile-control, $p = 0.079$). Metric data presented as boxplots with a center line as median, Tukey-style whiskers extend 1.5 times the interquartile range from $25^{th}$ and $75^{th}$ percentiles. Dots present individual datapoints. Data was analyzed with a linear mixed model. Sample size, MTRL: $n = 10$, control: $n = 20$; * = $p \leq 0.05$; ** = $p \leq 0.01$; *** = $p \leq 0.001$.

lesions. To this end, we used a virtual version of the MWM, a classic paradigm for testing memory-guided spatial navigation. The task was presented in either a stationary desktop setup with mainly visual input or a mobile immersive VR setup with multisensory input. Our results show that multisensory input modulated distinct aspects of memory-guided spatial navigation including spatial memory performance, navigation efficiency, and navigation strategies. Both, patients with hippocampal lesions and healthy control participants showed overall improvements in memory-guided spatial navigation when multisensory input was available. Remarkably, spatial memory performance improved more in patients than in control participants. In addition, the availability of multisensory information affected navigation strategies of patients with hippocampal lesions and control participants differently: whereas control participants employed more spatial landmarks to navigate to remembered locations, patients showed stronger replication of path sequences when they navigated freely compared to when they performed the same task in a stationary setup. Our results show that rearranged processing of multisensory input can efficiently compensate for hippocampal damage and should be taken into consideration when interpreting navigational behavior in human patients.

We observed improvements in spatial memory performance and navigation efficiency in humans with and without hippocampal lesions depending on the availability of multisensory input. Our results emphasize that multisensory input has direct implications for memory-guided spatial navigation performance. The observed behavioral changes can be explained by a modulation of neural activity across multiple brain areas including extrahippocampal brain regions. Indeed, functional magnetic resonance imaging studies have shown that the hippocampus and adjacent entorhinal and parahippocampal cortices are critical for processing spatial information during navigation[38,39]. However, these brain regions are part of a broader navigational network that extends beyond the medial-temporal lobe[1,13]. Within this network the hippocampus and adjacent structures interact and share computations for ego- and allocentric spatial representations with the striatum and neocortical brain areas such as the posterior parietal cortex and the retrosplenial cortex[1,40,41]. While the striatum contributes to stimulus-response learning, the retrosplenial cortex and the posterior parietal cortex integrate spatial information derived from head and body movements with visual spatial information based on landmarks which facilitates the localization of the self and familiar locations[42–44].

When participants navigate in the real world or in immersive VR, the visual system, vestibular organ, and proprioception relay congruent and complementary sensory information to the retrosplenial

cortex and the posterior parietal cortex[45]. This partially overlapping information from multimodal sources is then processed with low computational noise, which promotes the formation of robust spatial memory representations[46]. In contrast, when participants navigate virtual space projected on a desktop screen, body-based sensory input indicates that participants are stationary. These signals are at odds with the visual flow that simulates the experience of movement. As a result, different brain areas need to reconcile conflicting sensory information, leading to a greater cognitive demand for the generation of coherent spatial representations. These contradicting representations may alter the strategies for solving the navigation task and underlying neural processes[24,25], further disadvantaging participants in the stationary desktop setup compared to the mobile setup. Consequently, congruent sensory inputs to the posterior parietal cortex and retrosplenial cortex may have contributed to the improvements in performance observed in our study when body motion of participants was unrestricted and matched visual input.

Our study suggests that memory-guided spatial navigation in situations without multisensory input is more dependent on the integrity of the medial temporal lobe. The medial temporal lobe processes spatial features from visual flow independent of the viewpoint. For example, grid cells in entorhinal cortex extract the metric features of space, and the parahippocampal cortex processes information about landmarks even when self-motion is restricted[1,47–50]. This provides a contextual framework for spatial relations and enables processing of egocentric and allocentric representations in the hippocampus[40,51,52]. However, without complementary computations in extrahippocampal brain regions, the accuracy of the spatial representation relies on the accurate encoding of spatial information by the hippocampus. In this case, the influence of a hippocampal lesion becomes particularly evident. In humans, the hippocampal function has been shown to be lateralized, with the right hippocampus linked more to navigation-related functions compared to the left hippocampus[53,54]. The patients in our study had lesions in the right medial temporal lobe and they exclusively relied on the left medial temporal lobe for hippocampal computations. This resulted in discernible deficits in memory-guided spatial navigation especially in the stationary setup, with the absence of congruent body-based spatial representations hosted by extrahippocampal brain areas.

Our results further show that the availability of multisensory input compensates for deficits in memory-guided spatial navigation due to a hippocampal lesion. In the hippocampus, unique combinations of place cells encode specific locations in relation to environmental cues[12,15,55]. However, even when hippocampal place cells are dysfunctional or absent, navigation to a target

location in relation to an allocentric frame of reference is possible[30,56]. One explanation is that hippocampal dysfunction is compensated for by increased computations in neocortical brain regions, evidenced by an altered frontoparietal activation pattern correlating with behavior[57]. In the broader network for spatial navigation, the medial temporal lobe and the parietal areas are likely to communicate via the retrosplenial cortex[58]. It is thus plausible that parietal and retrosplenial areas are more heavily recruited in absence of right hippocampal input, when the participants perform navigation tasks. Such a shift in computations for mental representations may subsequently translate into behavioral changes. Specifically, redistribution of computation from the hippocampus to the retrosplenial cortex and posterior parietal cortex can alter the relative proportion of allocentric versus egocentric spatial representations. In the joint integrative processing of spatial information, it would be adaptive to put more weight onto the intact parietal or retrosplenial representations that are anchored to egocentric reference frames. On the behavioral level, this would be reflected in the change in preferred navigation strategies. Indeed, we observed that patients with hippocampal lesions showed an increased use of egocentric navigation strategies and used landmark information less intensively in response to multisensory input compared to controls.

It should be noted, however, that the modulation in spatial navigation observed in our patients cannot be attributed exclusively to the use of egocentric representations. Performance above chance level in our version of the water maze requires allocentric representations, specifically in the probe trials with varying starting locations. In this scenario, a complete reliance on egocentric strategies such as path replication would misguide the participant, as they are required to approach the target location from different viewpoints. One explanation for the above chance performance in these trials is that computations for forming the allocentric representations overlap in both hippocampus and neocortex[1,13]. With allocentric representations available, egocentric representations can be embedded in the correct environmental context, translating otherwise misleading egocentric coordinates into an allocentric frame of reference.

Our systematic comparison of memory-guided navigation in humans in a stationary and a mobile setup can explain some of the controversial results in previous navigation studies. In prior research, it has been observed that patients with hippocampal lesions showed profound impairments in spatial memory performance in stationary virtual navigation tasks[19]. Patients with acute or chronic hippocampal lesions were affected, and the effect was particularly pronounced when the lesion was located in the right medial temporal lobe[20,59,60]. In contrast to observations in stationary navigation tasks, patients with unilateral hippocampal lesions were able to solve a physical analog of the water maze as efficiently as healthy controls[30,61], and even patients with bilateral hippocampal lesions navigated a physical or immersive VR version of the water maze better than chance[30,56,62]. Our results indicate that these conflicting findings can—at least partially—be attributed to the relevant difference in task design, namely the degree of mobility and the availability of multisensory input.

The importance of mobility for studying navigation is further highlighted by the fact that most of our understanding on the neural underpinnings of memory-guided spatial navigation is derived from behavioral experiments in freely moving animals. Due to interspecies differences, results from rodent studies cannot be readily translated to humans. However, in humans, navigation is often assessed in an immobile supine, sitting, or standing position in electrophysiological, imaging, and lesion studies[63–65]. In contrast, higher degrees of mobility in immersive VR environments allow for a more ecological comparison of knowledge about navigation between animals and humans. Combined with advances in mobile brain imaging technology, such as high-density mobile electroencephalography, optically pumped magnetoencephalography, and intracranial leads, immersive VR environments provide the opportunity to study neural substrates during full-body movement[66–70]. These methodological approaches can help identify shifts in neural substrates in response to changing behavioral demands on spatial navigation in future studies.

Our study highlights the importance of considering contextual factors as modulators of spatial navigation. We observed that multisensory input has a profound impact on memory-guided spatial navigation. Behavioral patterns may change significantly in response to contextual factors. Since unrestricted movement is a key feature of natural spatial navigation, behavioral observations in non-mobile navigation studies should be interpreted carefully and mobility should be allowed whenever possible.

Beyond a better understanding of spatial navigation, humans with deficits in spatial navigation especially benefit from improved ecological validity in studies. This includes patients with neurodegenerative diseases, but also patients with acute lesions to the navigational network, e.g., due to stroke or inflammatory brain disorders. Deficits of navigational abilities in these patients could be treated by rehabilitation schemes that promote compensatory mechanisms in the navigational network.

In conclusion, the behavioral data from our experiment support the assumption of an extended large-scale navigation network in which brain regions continuously share the processing of egocentric and allocentric spatial representations rather than performing temporally and spatially separated computations in distinct neural substrates[1,13]. Complementary and redundant computations across brain regions allow for a flexible shift of processing according to current behavioral demands and reliability of spatial representations. Our results furthermore highlight the importance of considering contextual factors such as the availability of multisensory input in studies on memory-guided spatial navigation in patients with hippocampal damage and in healthy participants.

## Methods

**Participants**. In total, thirty-four participants took part in the experiment and thirty participants were included in the final dataset of our study (18 female, 12 male; Table 1). Eleven patients were recruited through our Department of Neurology who had undergone unilateral partial resection of the right medial temporal lobe (MTLR), including the hippocampus, due to hippocampal sclerosis and intractable epilepsy ($n = 7$) or due to removal of a benign tumor ($n = 4$) (Fig. 5, Table 2). The other inclusion criteria for patients were as follows: Age 18–65 years, fluent German (at least C1 level), postoperative neurological examination was normal, no other neuropsychiatric or severe internal diseases were reported, vision and hearing were normal or corrected to normal, no subjective memory complaints and navigation deficits in daily life were reported, and patients could be fully reintegrated into their personal and professional lives after surgery. Another requirement for inclusion in the study was a postoperative period of at least 6 months before the test to ensure sufficient recovery time after surgery. Each patient was matched with two healthy control participants in terms of gender, age, and education level. The control subjects were recruited via online advertising. Participants in the final data set were aged between 22 and 61 years, and four patients were taking anticonvulsant medication at the time of the study. Clinical cognitive assessment was not considered in the recruitment of patients, as patients with unilateral lesions are more likely to have subtle memory deficits that are not usually detected in routine

**Table 1 Participant data.**

|  | MTRL (n = 10) | Control (n = 20) | P-value |
|---|---|---|---|
| Female/male (count)[a] | 6/4 | 12/8 | 1.0[b] |
| Age (years) | 41 (22-61) | 41 (22-61) | 0.939[c] |
| Years of education | 16 (12-20) | 15.5 (12-19) | 0.607[c] |
| Santa Barbara Sense of Direction Scale | 4.1 (3.3-4.8) | 4.8 (3.3-6.1) | 0.031[c] |

Data presented as median, minimum, and maximum.
[a]All participants stated that their gender was the same as their sex.
[b]$\chi^2$-test.
[c]Mann-Whitney-U-test.

examinations[71]. One patient and three control participants were later excluded due to cyber sickness or an additional neuropsychiatric disorder that was not known at the time of the experiment. All participants provided written informed consent in accordance with the Declaration of Helsinki and all procedures were approved by the local ethics committee of Charité-Universitätsmedizin Berlin. All ethical regulations relevant to human research participants were followed.

**Lesion evaluation**. All patients except No. 5 and No. 9 participated in previous studies where lesion size was analyzed[72,73]. Nos. 5th and 9th lesions were additionally analyzed using MRI scans from clinical routine. Briefly, 47 coronal T1 sections of the whole brain with an individual thickness of 4 mm were used to determine individual lesion size. The extent of each lesion was determined from rostral to caudal using previously proposed landmarks[74–77].

**Experimental setup**. To investigate the influence of multisensory input on memory-guided spatial navigation, we tested spatial memory and navigation behavior in a virtual environment presented either on a screen on which participants navigated with a joystick (stationary) or in an immersive VR setup in which participants moved freely (mobile) (Fig. 1a, b).

The duration of the entire experiment varied between four and six hours, including technical preparations and breaks. We performed all experiments at the Berlin Mobile Brain/Body Imaging Labs (BeMoBIL) at the Technische Universität Berlin.

Participants performed the task equipped with a fully mobile EEG system (Supplementary Methods 2), the data from which will be reported in detail in a follow-up study with the focus on the electrophysiological dynamics during spatial navigation. EEG data will be analyzed to confirm or reject the hypotheses about brain dynamics raised in the current study.

**Virtual navigation task**. To investigate memory-guided spatial navigation, we used a modified virtual version of the MWM-task developed in Unity 3D (v.2018.4.13f1)[14,15,78]. The virtual environment consisted of an open, circular arena surrounded by environmental cues (Fig. 1c, d). The arena had a radius of 3.8 (virtual) meters and was bounded by a 1.7 (virtual) meter high wall. The ground inside the arena was covered with a half-transparent fog model. A skybox with clouds was rendered in the background and the arena was located in the valley of a hilly terrain. Three different buildings were placed in a triangular formation in the hilly landscape (Medieval house 3D, PBR medieval houses pack, Church model, Medieval village environment, Medieval castle pack available in Unity 3D store; Unity's standard Unity Asset Store End User License Agreement (EULA), extension asset). To avoid carryover effects between the stationary desktop setup and the mobile VR setup, we created two different versions of the virtual environment (Fig. 1c). The scene versions

and presentation order for the stationary setup and the mobile setup were matched between participants.

**Behavioral testing**. Six experimental blocks were presented in each of the stationary desktop and mobile VR setups (Fig. 1d, Supplementary Table 1). A block was defined by a set of spatial parameters, namely the start location and the target location. First, the six start locations were located on each end of the radial axes, equally dividing the circle into six areas (at 0, 60, 120, 180, 240, and 300 degrees). For each start location, the corresponding target location was located on one of the center axes of the four quadrants defined with respect to the start location (relative angles ± 45 degrees or ±135 degrees). The distance of the target location from the center was randomly sampled from a uniform distribution over the interval of [0.2, 0.8] × arena radius (3.8 (virtual) meters). The six sets of block-specific spatial parameters were generated once and used for all participants and both sessions (Supplementary Table 1). However, the order of the blocks was randomly permuted for each setup. At the target location a randomly selected object model (Toys Pack, Lowpoly Flowers, 3D Cute Toy Models available in Unity 3D store; Unity's standard Unity Asset Store End User License Agreement (EULA), extension asset) was used per block.

Each block started with three learning trials, followed by four probe trials. A disorientation task was inserted between every consecutive pair of trials. In learning trials, participants searched for the hidden target object in the arena. The object gradually appeared when approached (<1.2 (virtual) meters) and was registered as found when the participant was closer than 0.8 (virtual) meters. The target object remained visible for a maximum of 20 s, and the participant was instructed to remember its location. This phase could optionally be ended earlier by pressing a key.

In probe trials, participants were asked to navigate back to the remembered target location. While the target location remained fixed, the start location varied between the four probe trials. The start locations were defined as rotations of the origin—the start location used during learning—around the center by 0, 90, 180, and 270 degrees. The four rotations were presented in a randomly permuted order within a block. In these trials, the target object stayed invisible, and the participants completed the task by pressing the key after having positioned themselves at the remembered location of the target.

A disorientation task was inserted between all pairs of consecutive trials or after termination of a break between blocks in both the stationary and mobile sessions. This was to prevent participants from using a simplistic strategy of immediately backtracking the learned trajectory from the previous trial. In the disorientation task, all visual features that could be used as a spatial cue were hidden, including the skybox. Participants were first asked to navigate to a waypoint—a blue sphere—at the center of the arena. Then a white sphere appeared in the viewing direction, which guided the participant to turn their body following a sequence of three rotations. The rotation sequence was randomized between right-left-right and left-right-left. After following the sequence of rotation, they were asked to walk straight to the starting location of the next trial indicated by a waypoint. Only then the sky and other spatially relevant features in the virtual environment were revealed again and the next trial started. The reasoning behind this manipulation was that the representation of the location of oneself formed in a trial should be reset at the beginning of the next one. As it is physically challenging to teleport participants in real world, we have rotated the virtual environment and masked the potential dissonance with the disorientation task.

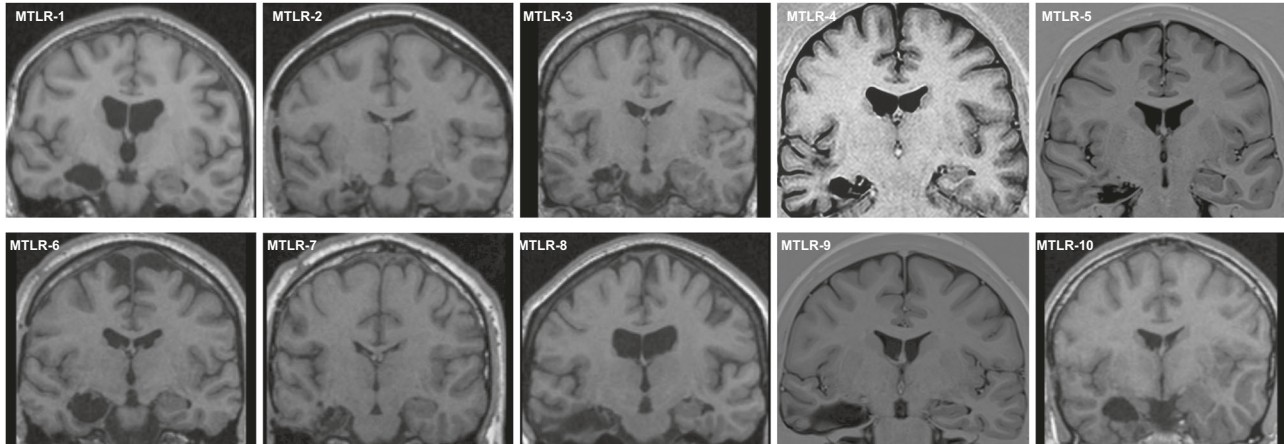

**Fig. 5 Example images of unilateral lesions of the medial temporal lobe.** Postoperative coronal T1 MRI images of the brain show the unilateral lesion of the medial temporal lobe including the right-sided hippocampus, while the left-sided hippocampus is intact.

**Table 2 Patient data and individual lesion extent.**

|  | Neuropathology | Postoperative time | Hip | ERC | PRC | PHC | ITC |
|---|---|---|---|---|---|---|---|
| MTLR-1 | Pilomyxoid astrocytoma | 15 years | + | + | + | 0 | 0 |
| MTLR-2 | Hippocampal sclerosis | 13 years | ++ | ++ | ++ | + | ++ |
| MTLR-3 | Hippocampal sclerosis | 14 years | ++ | ++ | ++ | 0 | ++ |
| MTLR-4 | Pilocytic astrocytoma | 18 years | ++ | ++ | ++ | + | ++ |
| MTLR-5 | Hippocampal sclerosis | 18 months | ++ | ++ | ++ | + | +++ |
| MTLR-6 | Neuroepithelial tumor | 14 years | + | ++ | ++ | 0 | 0 |
| MTLR-7 | Hippocampal sclerosis | 17 years | ++ | ++ | ++ | + | ++ |
| MTLR-8 | Hippocampal sclerosis | 17 years | ++ | ++ | ++ | + | +++ |
| MTLR-9 | Hippocampal sclerosis | 11 months | ++ | ++ | ++ | + | +++ |
| MTLR-10 | Pilocytic astrocytoma | 19 years | ++ | ++ | ++ | 0 | 0 |

*HIP hippocampus, ERC entorhinal cortex, PRC perirhinal cortex, PHC parahippocampal cortex, ITC inferior temporal cortex, VOL lesion volume (ml), 0 region unaffected, + rostrocaudal lesion extent ≤20 mm, ++ ≤40 mm +++ >40 mm.*

Prior behavioral testing, all participants became acquainted to the task requirements such as the mode of control by performing a baseline and a practice block before the start of the first experimental block in each setup. A baseline block consisted of a phase of 30 s where they stood still looking in a specific direction from within the arena, followed by a navigation task where they followed 36 waypoints appearing in the arena one after another. Per setup, the baseline block was presented three times: before the first experimental block, after the third block, and at the end of the sixth (last) block. The practice block was presented only once at the beginning of each setup by default and then demonstrated one learning trial and one probe trial with a disorientation task in between.

**Technical equipment: stationary desktop setup**. In the stationary setup, the virtual environment was presented with a first-person view on a wall-mounted screen (43 inches, 3840 × 2160) in the same room as the mobile VR setup. Participants viewed the screen while standing approximately 1.2 meters away and simulated movement using a joystick (Speedlink Dark Tornado) placed on a desk in front of them. The heights of both the screen and the desk were adjusted according to the height of the participant. To navigate in the virtual environment, participants rotated their perspective around the up-down axis (yaw) by tilting the joystick to the left or to the right. Likewise, forward and backward translation was controlled by tilting the joystick forward or backward. The speed of translation was 1.4 virtual meters per second. The rotation speed was 50 degrees per second. The time series of positions and orientation data of the virtual camera was sampled at 60 Hz (refresh rate of the display) and streamed to the Lab Streaming Layer[79]. Participants pressed a red button on the joystick to respond or to terminate breaks between blocks.

**Technical equipment: mobile VR setup**. In the immersive mobile VR setup, a virtual environment was presented to the participants using a head-mounted immersive VR display (HTC Vive Pro, 110 degrees field of view). To enable wireless navigation within the room, a wearable gaming PC (Zotac: PC Partner Limited), powered by portable batteries, was used to generate the graphical input to the HMD. The time series of positions and orientation data of the HMD were sampled at 90 Hz (refresh rate of the display) and were streamed via Wi-Fi to the Lab Streaming Layer on the recording PC[79]. The navigable area in the room was ~15 × 9 meters. However, participants were instructed to always stay within the boundary of the virtual arena (a walled circle with 3.8 meters radius). During the task, there were no external cues (sound or air flow) that may have informed participants of their position in the room. While performing the tasks, participants held an HTC Vive controller and pressed the trigger key to respond or to terminate breaks between blocks.

**Data analysis**. We recorded the participant's position in the virtual environment as x, y coordinates in a Cartesian coordinate

system along with a time stamp and rotations as quaternions. Yaw angles were computed by converting quaternions to Euler angles and the channel preprocessed with a 6 Hz low pass filter to capture only those motions that had a relevant time scale. We pre-processed the navigation data in MathWorks® Matlab (version 2021a). Using the position and yaw data, we computed different variables to capture distinct aspects of memory-guided spatial navigation. Here, we focused on spatial memory performance, navigation efficiency, and navigation strategies.

First, we assessed spatial memory performance by determining how well participants could remember the target locations in the test trials and how accurate and consistent the underlying spatial memory representations were. To this end, we computed the memory score and the scatter of final locations.

*Memory score* was chosen as a measure for spatial memory. For each trial, we calculated the Euclidean distance between the target location and the final location. We compared this value to a reference distribution obtained by calculating the Euclidean distance between each target location and 1000 randomly selected locations in the arena. The memory score corresponds to the percent rank of the distribution, i.e., the proportion of randomly selected locations that were farther from the target location than the final location. Thus, the memory score ranges from zero to 100%, with 100% representing a perfect recall rate and 50% representing randomness. A value between 50% and 0 indicates a systematic bias in the false direction[28,29].

$$\text{Euclidean distance} = \sqrt{\left(x_{final\ location} - x_{target\ location}\right)^2 + \left(y_{final\ location} - y_{target\ location}\right)^2}$$

$$\text{Memory score} = 100 - proportion\ of\ random\ locations\ with\ less\ distance$$
$$to\ target\ location\ than\ the\ final\ location$$

*Scatter of final locations* was chosen as a measure for spatial precision which also depends on hippocampal integrity[30,31]. Regardless of the distance to the actual target location, precision indicates how consistent responses were per target location. The scatter of participants' responses was calculated as the average distance between each of the four final locations per object location. A smaller scatter meant higher spatial precision (Fig. 2c).

$$\text{Avg. distance to final location} = \frac{\sum_{\substack{i=1 \\ j=1}}^{c(n,2)} \sqrt{\left(x_i - x_j\right)^2 + \left(y_i - y_j\right)^2}}{c(n,2)}$$

Second, we evaluated temporal and spatial navigation efficiency. The temporal efficiency describes how quickly the final location was reached, while the spatial efficiency describes how directly the final location was reached.

*Latency to final location* was determined as measure for temporal efficiency. We calculated the latency to final location by subtracting the time of the trial onset from the time of the trial offset, in seconds.

$$\text{Latency to final location} = time_{offset} - time_{onset}$$

*Path error* was computed as first measure for spatial efficiency. Path error describes the directness of the participant's path to the final location. Here, we calculated the length of the participant's path and the ideal path length to the final location (ideal path is defined as the straight line from the start location to the final location, and its length corresponds to the Euclidean distance between two locations). We subtracted the ideal path length from the measured path length to obtain the excess path length. The excess path length was divided by the ideal path length and finally multiplied by 100, to yield the path

error. The path error ranges from zero to infinity, with higher values representing less direct paths.

$$\text{(Ideal) path length} = \sum_{i=1}^{n} \sqrt{\left(x_i - x_{i+1}\right)^2 + \left(y_i - y_{i+1}\right)^2}$$

$$\text{Path error} = \frac{path\ length_{actual} - path\ length_{ideal}}{path\ length_{ideal}} * 100$$

*Surface coverage* was used as second measure for spatial efficiency. Surface coverage refers to the maximum proportion of the arena surface visited by participants and provides an estimate of detours and the amount of target oriented spatial search during navigation. We calculated the difference between the minimum and maximum x and y coordinates, respectively, and determined the area of an ellipse to obtain an estimate of the area covered by the participants. We divided this value by the actual area of the arena to obtain the proportional amount of area covered. Higher values for covered area represent low spatial efficiency.

$$\text{Surface coverage} = \frac{pi * \left(abs(x_{min} - x_{max}) * (abs(y_{min} - y_{max})\right)}{pi * radius^2} * 100$$

Third, we assessed the navigation strategies underlying participants' performance. To this end, we used the observed movement patterns, such as the shape of the path to a location as well as the rotational behavior of the navigators, to infer the underlying navigation strategies. We distinguished three different parameter that reflect strategies that participants employed to find the target in the water maze: search accuracy, landmark use, and path replication. The choice of one strategy does not preclude the use of other strategies, as participants may switch between strategies and use more than one strategy simultaneously on the way to the target location[21,32].

*Average distance to final location* was computed as a measure for search accuracy, which describes the preferred spatial focus of the search behavior in the water maze. The focus of a search can be at the start location, at the final location, in the middle of the arena or randomly distributed in the arena. The focused location then has a higher-than-average number of coordinate points. A lower average distance reflects a preference for a more intense and focused search for the object near the final location, while a higher average distance is found when participants search mainly randomly or far away from the final location[33,34].

$$\text{Avg. distance to final location} = \frac{\sum_{i=1}^{n} \sqrt{\left(x_i - x_{final\ location}\right)^2 \left(y_i - y_{final\ location}\right)^2}}{n}$$

*Initial angular velocity* was evaluated as measure for landmark use. Landmark use describes the degree of visual exploration of the environment containing landmarks. The use of landmarks is the most purposeful strategic behavior in an allocentric spatial navigation task such as the water maze[15,19]. Efficient acquisition of information from the environment, such as landmarks, at the beginning of navigation accelerates self-localization, localization of the target location, and finally computation of the optimal path. We used integrated absolute angular velocity (idPhi) to quantify the extent to which participants used information from the environment. idPhi is derived from the heading data by unwrapping the yaw angles and taking the derivatives to calculate angular velocity. The instantaneous angular velocity values were averaged over the time window of interest to represent how much the participant had turned their head laterally. We chose the first five seconds as the time window of interest, as we were particularly interested in the exploration of the environment at the beginning of each trial. idPhi is commonly used as an index of vicarious trial and-error behavior and is known to be affected by

impairments of the medial temporal lobe[35,36,80].

$$\text{Initial angular velocity} = \frac{\sum_{i=1}^{n} \left| (yaw_i - yaw_{i+1}) \right|}{n}$$

*Trajectory distance* was determined as measure for path replication. Path replication describes the repetition of previously learned paths or path elements. This behavior requires route-based learning, which is realized by repeatedly navigating to the location of an object from a fixed location in learning trials. The use of repeated path sequences to reach the final location relies on egocentric representations, even when approaching the target location from a new start location[18,21,32,37]. The extent of repetition is reflected in the distance between the aligned trajectories of the last learning trial and the trajectories for each probe trial. We aligned the trajectory of the final learning trial with the trajectory of each probe trial by first rotating both trajectories around the center to obtain the same start coordinates, and then aligning the trajectories regardless of the actual path length using Matlab's dynamic time warping function *dtw*. To normalize the trajectory distance considering the number of data points, the *dtw-distance* was divided by the smallest number of matrix cells to be visited[81]. A smaller distance between matching trajectories reflects a greater repetition of the path or its elements, while large distances represent dissimilar trajectories.

$$\text{Trajectory distance} = \sqrt{\frac{dtw(A, B),'squared'}{\max(m, n)}}$$

**Statistics and reproducibility**. We performed the statistical analysis in R (v. 3.5). To determine whether our behavioral data met the assumption of a normal distribution, we applied the Shapiro-Wilk test. If the assumption of normal distribution was violated, we assessed the skewness and kurtosis of the data and applied a log transformation if the skewness was less than $-2$ or greater than 2 or the kurtosis was $<-7$ or $>7$, respectively.

Because our dataset consisted of consecutive measurements in two different experimental setups, we analyzed our data using a linear mixed model for two-sided testing and designed with the R package *lme4* (v.1.1-35),[82,83]. Fixed effects were group (between participants factor with two levels: MTLR and control) and setup (within-participants factor with two levels: stationary and mobile), and model covariates included session order, participant sex, age, and years of education, and random effects included participant ID to account for interindividual differences. The model was estimated using the restricted maximum likelihood method and degrees of freedom were calculated using the Satterthwaite method[84]. In case the main analysis revealed a significant interaction effect, a post-hoc test was performed using the R package *emmeans* with the Holm-Bonferroni correction for multiple comparisons to prevent an increase in type-I-errors (v.1.8.5),[85]. The R package *effectsize* was used to calculate effect sizes as Omega squared ($\omega^2$) (v.0.8.5).

To ensure comparable group characteristics with respect to sex age, and years of education, we used either the $\chi$2-independence test for nominal variables or the nonparametric Kruskal-Wallis test for metric variables. For all statistical tests applied, we set the significance level to the conventional level of 0.05.

The sample size for all statistical tests was as follows: MTLR, $n = 10$; control, $n = 20$. We provide two tables of results for learning trials and probe trials respectively (Supplementary Tables 2, 3). Data are presented as mean ± s.e.m. and 95% confidence interval. For the effect of the session order on the experimental variables, another table of results is provided with test statistics, *p* value, and effect size (supplementary tables 4).

The data are presented as box-and-whisker plots with a center line representing the median and with individual data points overlaid to show the full data distribution.

**Reporting summary**. Further information on research design is available in the Nature Portfolio Reporting Summary linked to this article.

## Data availability

The data that support the findings of this study are available at the Open Science Framework (osf) at: https://osf.io/u47mj/, (unique identifier: https://doi.org/10.17605/OSF.IO/U47MJ).

## Code availability

Our matlab- and R-functions are available at the Open Science Framework (osf) at: https://osf.io/u47mj/, (unique identifier: https://doi.org/10.17605/OSF.IO/U47MJ). The software for the virtual water maze and the acquisition of trajectories is available upon reasonable request from the corresponding author.

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

## Acknowledgements

We thank Tore Knabe for programming the water maze task. We thank Timo Berg for technical assistance. This study was funded by the Deutsche Forschungsgemeinschaft DFG, German Research Foundation—Project number 327654276—SFB 1315.

## Author contributions

Conceptualization: D.I., S.J., K.G., C.J.P., and C.F.; Methodology: D.I., S.J., K.G.; Participant recruitment: D.I., C.J.P., and C.F.; Data acquisition: D.I., S.J., and P.M.M.; Data analysis: D.I. and S.J.; Statistical analysis: D.I. and S.J.; Visualization: D.I.; Supervision: C.J.P., K.G., and C.F.; Writing—original draft: D.I., S.J., and C.F.; Writing—review & editing: D.I., S.J., P.M.M., K.G., C.J.P., and C.F.; Technical equipment: K.G.; Funding: C.J.P. and C.F. All authors approved the final manuscript.

## Funding

## Competing interests

The authors declare no competing interests.
