## [Peer review file · Communications Biology]

Reviewers' comments:

Reviewer #1 (Remarks to the Author):

In "Multisensory input modulates memory-guided spatial navigation", Iggena and Jung et. al present data from thirty participants, including ten with unilateral partial resection of the right medial temporal lobe (MTL). Participants performed stationary and innovative mobile experimental tasks. Authors report that participants with MTL resections benefit from multisensory input in the mobile study; this benefit was relatively larger than in control participants with intact MTL.

These and other results presented are novel and complement the literature very well. Yet, I do think that the comments below could contribute to improving the manuscript before publication.

Major:

A.)

More Information on the inclusion criteria/ recruitment of patients would be helpful; MRI showing their resection would be insightful if available. Other questions understanding the included population here would be:

Was the clinical assessment of memory performance considered when recruiting?

Did participants have memory or navigation complaints?

B.) Mention if multiple comparison corrections, wherever appropriate, were applied or not.

Minor:

Abstract:

"However, in human navigation studies, the full range of sensory information is often unavailable."

Explain why that is the case

"Our results show that availability of multisensory input..." This sentence is quite hard to read; consider splitting it up in two sentences

Results:

Pg 3: Although MWM is a standard paradigm, briefly explain this task here, mentioning that participants were not in the water. So it is an adapted MWM for humans.

Explicitly mention how many participants were included in the second paragraph.

Pg.6: "Both, participants..." add comma; otherwise, one might think there are two participants in the study.

MTLR: Although the methods define the abbreviation, please define it before mentioning it in text and captions for more clarity.

Scatter: "average of all six distances"; relative to each other would be helpful for better understanding.

Fig.2: 1000 locations, following which distribution – spatial unit distribution

Pg 7: "reduction in average latency"; relative to the first one?

Discussion:

Both, patients – insert a comma.

The rest reads well to me!

Reviewer #2 (Remarks to the Author):

Reviews for "Multisensory input modulates memory-guided spatial navigation" Communications Biology, COMMSBIO-23-1937-T

The current study investigated whether multisensory input improves spatial memory in patients with hippocampus lesions. Patients and normal controls were tested in a Morris Water Maze task in desktop VR with a joystick and in immersive VR with an HMD. They navigated to learn target locations and later were tested to find the targets without feedback. Compared with the desktop

VR condition, in immersive VR, both groups showed improvement in their performance while the patient group showed a greater effect. These results suggest that multisensory input may facilitate the information processing in extrahippocampal areas and compensate for the hippocampus lesions.

Overall, the current study clearly presented the study. The research question and the hypotheses are well stated in the manuscript. I would like to see the paper published in the end. I have a few suggestions that the authors may consider to improve the manuscript.

Major comments:

1. Figure 1a: These pictures show participants wearing mobile EEG recording devices. But the manuscript did not mention anything about that except in the last section of the supplementary materials. I would suggest to clarify this in the main text with reasons for why EEG recordings were in this study but not reported in the paper.
2. Methods: Did the participants perform any disorientation between trials in the desktop condition? I did not see this information in the methods. Please add a little more description.
3. If I understand it correctly, the two conditions, desktop vs. immersive VR, were counterbalanced in order. If there is no order effect, please clarify this in the text.

Minor comments:

4. I am not sure if this is a software issue but all "omega squared" signs for statistics in the pdf did not appear appropriately. I would also like to see how this effect size statistic was calculated.
5. Page 22, Line 671: There is a typo "finalt" which should be "final".
6. The font size in the figures were a little too small.

Reply to the Reviewers

We would like to thank the Editor and the Reviewers for taking their time to thoroughly read and review our manuscript. We are grateful for their detailed and constructive comments and have made every effort to meet the reviewers' criticisms, which we feel has significantly improved our manuscript.

#Reviewer 1

We thank the reviewer for the evaluation and are grateful for all suggestions that we address in our response point by point below.

1. More Information on the inclusion criteria/ recruitment of patients would be helpful;

We thank the reviewer for this question and agree that we could have presented the inclusion criteria more precisely. Patients were recruited through our own neurology clinic, which works closely with the Königin-Elisabeth Herzberge Hospital in Berlin, whose neurology clinic specializes in epilepsy diagnostics and performs the preoperative work-up and postoperative treatment of patients with epilepsy-related temporal lobe resection. The inclusion of patients was performed as follows: Patients of both sexes aged 18 to 65 years were included if the unilateral medial temporal lobe resection was right-sided and included the hippocampus, the postoperative neurological examination was normal, vision and hearing were normal or corrected to normal, no subjective memory complaints and navigation deficits in daily life were reported, and patients could be fully reintegrated into their personal and professional lives after surgery. Another requirement for inclusion in the study was a period of at least 6 months until the test to ensure sufficient recovery time after surgery. One patient and three control participants were later excluded due to cyber sickness or an additional neuropsychiatric disorder that was not known at the time of the experiment.

To ensure a better understanding of the patient sample, we have changed the description of the participants in the Methods section.

The following paragraph

Methods, p. 18, "Participants", ll. 498 - 510: We included a total of thirty participants in our study (18 female, 12 male; table 1). Ten participants were patients who had undergone unilateral partial resection of the right medial temporal lobe (MTLR), including the hippocampus (Table 2). Resections were performed either because of hippocampal sclerosis and intractable epilepsy ($N = 6$) or because of removal of a benign tumor ($N = 4$). At the time of the study, four patients were taking anticonvulsant medication. None of the ten patients reported an additional neuropsychiatric disorder. All patients were fully integrated in their preoperative social and professional lives. We recruited the patients through the Department of Neurology at Charité-Universitätsmedizin Berlin. Each patient was matched with two healthy control participants for sex, age, and education level. Control subjects were recruited via online advertising.

Participants were between 22 and 61 years old, spoke German fluently, had normal or corrected-to-normal vision as well as normal hearing. All participants gave written informed consent according to the declaration of Helsinki, and all procedures were approved by the local ethics committee of Charité-Universitätsmedizin Berlin.

was exchanged with

Methods, p. 19, “Participants”, ll. 537 - 557: “In total, thirty-four participants took part in the experiment and thirty participants were included in the final dataset of our study (18 female, 12 male; table 1). Eleven patients were recruited through our Department of Neurology who had undergone unilateral partial resection of the right medial temporal lobe (MTLR), including the hippocampus, due to hippocampal sclerosis and intractable epilepsy (n = 7) or due to removal of a benign tumour (n = 4) (Fig.5, Table 2). The other inclusion criteria for patients were as follows: Age 18 to 65 years, fluent German (at least C1 level), postoperative neurological examination was normal, no other neuropsychiatric or severe internal diseases were reported, vision and hearing were normal or corrected to normal, no subjective memory complaints and navigation deficits in daily life were reported, and patients could be fully reintegrated into their personal and professional lives after surgery. Another requirement for inclusion in the study was a postoperative period of at least 6 months before the test to ensure sufficient recovery time after surgery. Each patient was matched with two healthy control participants in terms of gender, age, and education level. The control subjects were recruited via online advertising. Participants in the final data set were aged between 22 and 61 years, and four patients were taking anticonvulsant medication at the time of the study. Clinical cognitive assessment was not considered in the recruitment of patients, as patients with unilateral lesions are more likely to have subtle memory deficits that are not usually detected in routine examinations (Esfahani-Bayerl et al., 2016). One patient and three control participants were later excluded due to cyber sickness or an additional neuropsychiatric disorder that was not known at the time of the experiment. All participants provided written informed consent in accordance with the Declaration of Helsinki and all procedures were approved by the local ethics committee of Charité-Universitätsmedizin Berlin.”

2. MRI showing their resection would be insightful if available.

We thank the reviewer for the valuable suggestion and added the following image to the methods section, ll. 565 – 568:

“Fig. 5. Example images of unilateral lesions of the medial temporal lobe.

Postoperative coronal T1 MRI images of the brain show the unilateral lesion of the medial temporal lobe including the right-sided hippocampus, while the left-sided hippocampus is intact.”

Additionally, we clarified the lesion size in the table which describes the postsurgical lesions in more details:

The following table, p. 34

Table 2. Patient data and individual lesion extent.

	Neuropathology	Postoperative time					
			Hip	ERC	PRC	PHC	ITC
MTLR-1	Piloxyoid astrocytoma	15 years	+	+	+	0	0
MTLR-2	Hippocampal sclerosis	13 years	++	++	++	+	++
MTLR-3	Hippocampal sclerosis	14 years	++	++	++	0	++
MTLR-4	Pilocytic astrocytoma	18 years	++	++	++	+	++
MTLR-5	Hippocampal sclerosis	18 months	++	++	++	+	+++
MTLR-6	Neuroepithelial tumor	14 years	+	++	++	0	0
MTLR-7	Hippocampal sclerosis	17 years	++	++	++	+	++
MTLR-8	Hippocampal sclerosis	17 years	++	++	++	+	+++
MTLR-9	Hippocampal sclerosis	11 months	++	++	++	+	+++
MTLR-10	Pilocytic astrocytoma	19 years	++	++	++	0	0

was exchanged with the following table, p. 36:

Table 2. Patient data and individual lesion extent.

	Neuropathology	Postoperative time	Hip	ERC	PRC	PHC	ITC
MTLR-1	Pilomyxoid astrocytoma	15 years	+	+	+	0	0
MTLR-2	Hippocampal sclerosis	13 years	++	++	++	+	++
MTLR-3	Hippocampal sclerosis	14 years	++	++	++	0	++
MTLR-4	Pilocytic astrocytoma	18 years	++	++	++	+	++
MTLR-5	Hippocampal sclerosis	18 months	++	++	++	+	+++
MTLR-6	Neuroepithelial tumor	14 years	+	++	++	0	0
MTLR-7	Hippocampal sclerosis	17 years	++	++	++	+	++
MTLR-8	Hippocampal sclerosis	17 years	++	++	++	+	+++
MTLR-9	Hippocampal sclerosis	11 months	++	++	++	+	+++
MTLR-10	Pilocytic astrocytoma	19 years	++	++	++	0	0

HIP, hippocampus; ERC, entorhinal cortex; PRC, perirhinal cortex; PHC, parahippocampal cortex; ITC, inferior temporal cortex; VOL, lesion volume (ml); 0, region unaffected; +, rostrocaudal lesion extent \leq 20 mm; ++, \leq 40 mm; +++, $>$ 40 mm.

All patients except No. 5 and No. 9 participated in previous studies analysing lesion size (Braun et al., 2008, Braun et al., 2011). Nos. 5th & 9th were analysed similarly to Braun et al., 2008. Briefly, 47 coronal T1 sections of the whole brain with an individual thickness of 3.5 mm were used to determine individual lesion size. The extent of each lesion was then determined from rostral to caudal using the landmarks proposed by Insausti et al. (1995, 1998), Insausti and Amaral (2004) and derived by Mai et al. (2004).

We added the following paragraph to the methods section, p. 19, ll. 559 – 564:

“Lesion evaluation

All patients except No. 5 and No. 9 participated in previous studies where lesion size was analysed (Braun et al., 2008, Braun et al., 2011). Nos. 5th & 9th lesions were analysed similarly as previously described

with MRI scans from clinical routine. Briefly, 47 coronal T1 sections of the whole brain with an individual thickness of 4 mm were used to determine individual lesion size. The extent of each lesion was determined from rostral to caudal using previously proposed landmarks (Insausti et al., 1995, 1998, 2004, Mai et al., 2004).”

References:

Braun M, Weinrich C, Finke C, Ostendorf F, Lehmann TN, Ploner CJ. Lesions affecting the right hippocampal formation differentially impair short-term memory of spatial and nonspatial associations. *Hippocampus*. 2011 Mar;21(3):309-18. doi: 10.1002/hipo.20752. PMID: 20082291.

Braun M, Finke C, Ostendorf F, Lehmann TN, Hoffmann KT, Ploner CJ. Reorganization of associative memory in humans with long-standing hippocampal damage. *Brain*. 2008 Oct;131(Pt 10):2742-50. doi: 10.1093/brain/awn191. Epub 2008 Aug 29. PMID: 18757465.

Insausti R, Juottonen K, Soininen H, Insausti AM, Partanen K, Vainio P, et al. MR volumetric analysis of the human entorhinal, perirhinal, and temporopolar cortices. *Am J Neuroradiol* 1998; 19: 659–71

Insausti R, Amaral DG. Hippocampal formation. In: Paxinos G, Mai J, editors. *The human nervous system*. Amsterdam: Elsevier Academic Press; 2004. p. 871–915.

Mai J, Assheuer J, Paxinos G. *Atlas of the human brain*. Amsterdam: Elsevier Academic Press; 2004.

3. Other questions understanding the included population here would be: Was the clinical assessment of memory performance considered when recruiting?

We thank the reviewer for this question. Clinical cognitive assessment is routinely performed before and after resection of the medial temporal lobe, but memory performance in this clinical assessment was not considered in the current recruitment and we did not pre-select based on memory performance in other standardized clinical tests. Since eight out of ten patients – all but no. 5 and 9 – have already participated in previous published studies of our research group, we know that the patients are more likely to have mild memory deficits that can be detected by more specific tests outside of routine clinical practice, as shown in previous studies of our research group.

For example, we did not find differences for immediate and delayed recall of the complex Rey figure when patients with unilateral medial temporal lobe resections were compared to healthy controls. In contrast, to patients with bilateral lesions (see table).

Table 2
Socio-demographic and neuropsychological data of patient and control groups.

Group	Age	YOE	TSL	MWT (Verbal IQ)	LPS (Nonverbal IQ)	BTfw	BTbw	ROCF copy	ROCF immediate	ROCF delayed
R-MTL	27.3 ± 3.4	13.7 ± 0.8	32.4 ± 7.9	107.2 ± 4.5	31.2 ± 3.8	8.7 ± 0.4	8.5 ± 0.4	34.5 ± 0.6	19.8 ± 1.1	19.4 ± 1.4
L-MTL	37.8 ± 4.7	16.1 ± 1.0	45.0 ± 22.5	103.2 ± 5.3	29.0 ± 1.9	8.4 ± 0.7	8.0 ± 0.3	34.8 ± 0.5	26.1 ± 3.2	25.5 ± 3.4
Con 1	34.5 ± 4.8	14.1 ± 0.5	n.a.	109.0 ± 3.4	26.8 ± 1.6	9.0 ± 0.8	7.9 ± 0.2	35.3 ± 0.3	22.4 ± 2.0	22.9 ± 1.7
p value*	0.44	0.08	0.89	0.67	0.48	0.72	0.44	0.41	0.43	0.35
HSE	52.1 ± 4.6	17.0 ± 1.5	37.4 ± 13.4	105.8 ± 3.0	22.5 ± 2.0	6.9 ± 0.4	6.6 ± 0.5	35.1 ± 0.3	19.2 ± 3.0	16.5 ± 2.3
GCH	54.1 ± 4.3	13.6 ± 0.6	26.7 ± 8.3	114.1 ± 5.9	22.7 ± 2.2	7.1 ± 0.6	6.7 ± 0.5	35.4 ± 0.2	24.7 ± 1.9	24.1 ± 2.2
Con 2	48.1 ± 3.2	16.4 ± 1.1	n.a.	119.2 ± 3.9	27.4 ± 1.9	7.8 ± 0.7	7.3 ± 0.5	35.6 ± 0.3	25.6 ± 2.4	25.9 ± 2.2
p value*	0.34	0.12	0.91	0.06	0.29	0.33	0.36	0.42	0.28	0.03

BTfw, block tapping forward; BTbw, block tapping backward; IQ, intelligence quotient; LPS, Leistungsprüfsystem (German equivalent to Raven's Progressive Matrices); MWT, Mehrfachwahl-Wortschatztest (German equivalent to the National Adult Reading Test); n.a., not applicable; ROCF, Rey–Osterrieth Complex Figure Task; TSL, time since lesion (months); YOE, years of education.

* Kruskal–Wallis-test; group averages are means ± SEM.

We added the following sentence to the methods section, p.18, “participants”, ll. 552 - 554:

“Clinical cognitive assessment was not considered in the recruitment of patients, as patients with unilateral lesions are more likely to have subtle memory deficits that are not usually detected in routine examinations (Esfahani-Bayerl et al., 2016).”

Reference:

Esfahani-Bayerl N, Finke C, Braun M, Düzel E, Heekeren HR, Holtkamp M, Hasper D, Storm C, Ploner CJ. Visuo-spatial memory deficits following medial temporal lobe damage: A comparison of three patient groups. *Neuropsychologia*. 2016 Jan 29;81:168-179. doi: 10.1016/j.neuropsychologia.2015.12.024. Epub 2016 Jan 5. PMID: 26765639.

4. Did participants have memory or navigation complaints?

Prior to the experiment, participants reported no memory or navigation deficits that affected their daily personal or professional lives during telephone screening. However, patients scored lower on the Santa Barbara Sense of direction Scale (SBSOD) compared to healthy controls, indicating lower spatial navigational ability in general. We have now included the results in Table 1 in the manuscript.

The following table, p. 33:

Table 1. Participant data. Data presented as medium ± SEM, minimum, and maximum.

	MTRL (n = 10)	Control (n = 20)	P-value
Female/male (count)	6/ 4	12/ 8	1.0 ^a
Age (years)	41 (22 – 61)	41 (22 – 61)	0.939 ^b
Years of education	16 (12 – 20)	15.5 (12 – 19)	0.607 ^b

^a χ^2 -test, ^b Mann-Whitney-U-test

was exchanged with, p. 35:

Table 1. Participant data. Data presented as medium ± SEM, minimum, and maximum.

	MTRL (n = 10)	Control (n = 20)	P-value
Female/male (count)	6/ 4	12/ 8	1.0 ^a
Age (years)	41 (22 – 61)	41 (22 – 61)	0.939 ^b
Years of education	16 (12 – 20)	15.5 (12 – 19)	0.607 ^b

Santa Barbara Sense of Direction Scale	4.1 (3.3 – 4.8)	4.8 (3.3 – 6.1)	0.031 ^b
--	-----------------	-----------------	--------------------

^a χ^2 -test, ^b Mann-Whitney-U-test

Reference:

Hegarty et al., Development of a self-report measure of environmental spatial ability. *Intelligence*, 2002. [https://doi.org/10.1016/S0160-2896\(02\)00116-2](https://doi.org/10.1016/S0160-2896(02)00116-2)

5. Mention if multiple comparison corrections, wherever appropriate, were applied or not.

We thank the reviewer for pointing out the importance of multiple comparison corrections. Indeed, we applied corrections when the main analysis revealed a significant interaction of setup and group. The main analysis was followed by a Holm-Bonferroni correction. Compared to the standard Bonferroni correction, the Holm-Bonferroni correction is more powerful and less conservative when correcting for multiple comparisons.

We now describe the correction in more detail in the methods section in the paragraph Statistical analysis. Here we also provide information about the R-package used.

Methods, p. 23, “Statistical analysis”, ll. 715 - 717: “In case the main analysis revealed a significant interaction effect, a post-hoc test was performed using the R package *emmeans* with the Holm-Bonferroni correction for multiple comparisons (v.1.8.5),⁷⁷. “

was changed to

Methods, p. 26, “Statistics and Reproducibility”, ll. 781 - 784: “In case the main analysis revealed a significant interaction effect, a post-hoc test was performed using the R package *emmeans* with the Holm-Bonferroni correction for multiple comparisons to prevent an increase in type-I-errors.”

To highlight the application of a correction, we included a more detailed description in the table of results in the supplementary material 1.4 and 1.5, pp. 4-5:

The following description, p.4:

“**Supplementary table 2.** Table of results for learning trials. Data presented as mean \pm s.e.m and 95%-confidence interval.”

was changed to, p. 4:

“**Supplementary table 2.** Table of results for learning trials. Data presented as mean \pm s.e.m and 95%-confidence interval. In case main analysis revealed an interaction effect, post-hoc tests were performed with the Holm-Bonferroni-correction.”

and

“**Supplementary table 3.** Table of results for probe trials. Data presented as mean \pm s.e.m. and 95%-confidence interval.”

was changed to, p. 5:

“Supplementary table 3. Table of results for probe trials. Data presented as mean \pm s.e.m. and 95%-confidence interval. In case main analysis revealed an interaction effect, post-hoc tests were performed with the Holm-Bonferroni-correction.”

Furthermore, we now uploaded the R-functions used for statistical analysis to the online repository: <https://osf.io/u47mj/>

6. Abstract: “However, in human navigation studies, the full range of sensory information is often unavailable.” Explain why that is the case

We thank the reviewer for the suggestion to elaborate on the reason for the lack of sensory information in human studies. In humans, navigation is usually studied with 2D & 3D games using a joystick for navigation. Especially in imaging studies, such as MRI studies, movement is not possible, so participants do not receive sensory input through body movements. This is a methodological limitation that is not easily solved but can have an impact on navigation behaviour and prevent the recognition of underlying neural substrates.

To make this point clearer in the abstract, we have changed the following sentence:

Abstract, p. 2, ll. 48 - 49: “However, in human navigation studies, the full range of sensory information is often unavailable.”

to

Abstract, p.2, ll. 52 – 54: “However, in human navigation studies, the full range of sensory information is often unavailable due to the stationarity of experimental setups.”

Due to word limitations, we cannot provide further details in the abstract, but address this topic in the introduction.

7. “Our results show that availability of multisensory input...” This sentence is quite hard to read; consider splitting it up in two sentences

We agree and have, as suggested, split up this sentence that now reads:

Abstract, p.2, ll. 57 - 60: “Our results show that availability of multisensory input improves memory-guided spatial navigation in both groups. It has distinct effects on navigational behaviour, with greater improvement in spatial memory performance in patients.”

8. Results: Pg 3: Although MWM is a standard paradigm, briefly explain this task here, mentioning that participants were not in the water. So it is an adapted MWM for humans.

We thank you for this helpful hint. Accordingly, we added the following sentence

Results, p.4, 1st paragraph, ll. 114 – 116: “While a “water maze”, is typically implemented in rodent studies in a pool filled with water, we built a dry version of human scale water maze, using a virtual circular enclosure filled with virtual ground fog.”

9. Explicitly mention how many participants were included in the second paragraph.

As suggested, we changed the following sentence

Results, p.4, 2nd paragraph, ll. 116 – 117: “Patients with hippocampal lesions and their healthy controls learned the locations of objects by exploring the water maze, repeatedly starting from the same location.”

to

Results, p.4, 2nd paragraph, ll. 124 – 125: “Patients with hippocampal lesions (MTLR, n = 10) and their healthy controls (Control, n = 20) learned the locations of objects by exploring the water maze, repeatedly starting from the same location.”

10. Pg.6: “Both, participants...” add comma; otherwise, one might think there are two participants in the study.

As suggested, we inserted a comma in the following sentence and other comparable sentences. The above sentence has been changed to

Results, p.6, ll. 167 - 168: “Both, patients with medial temporal lobe resections (MTLR) including the hippocampus and healthy controls, performed above chance level in both the stationary desktop and the mobile VR setup.”

Another comma was inserted in the following sentence:

Results, p.8, ll. 220 - 221: “We found that temporal efficiency increased in both, patients and healthy controls when multisensory input was available in the mobile VR setup.”

Another comma was inserted in the following sentence:

Results, p.8, ll. 234 - 236: “As with temporal efficiency, we found that spatial efficiency increased in both, patients and healthy controls when multisensory input was available in the mobile VR setup. “

Another comma was inserted in the following sentence:

Discussion, p.15, ll. 394 - 396: “Both, patients with hippocampal lesions and healthy control participants showed overall improvements in memory-guided spatial navigation when multisensory input was available.”

11. MTLR: Although the methods define the abbreviation, please define it before mentioning it in text and captions for more clarity.

To clarify the abbreviations for the groups, the definitions are given earlier in the text and in the figure legend of figure two.

We changed the following sentence

Results, p.4, 2nd paragraph, ll. 116 – 117: “Patients with hippocampal lesions and their healthy controls learned the locations of objects by exploring the water maze, repeatedly starting from the same location.”

to

Results, p.4, 2nd paragraph, ll. 124 – 125: “Patients with hippocampal lesions (MTLR, n = 10) and their healthy controls (Control, n = 20) learned the locations of objects by exploring the water maze, repeatedly starting from the same location.”

Additionally, we changed the figure legend of figure two. The following sentence

Results, p. 8, ll. 197 – 198: “Sample size, medial temporal lobe resection group: $N = 10$, control: $N = 20$; * = $P \leq 0.05$; ** = $P \leq 0.01$; *** = $P \leq 0.001$.”

Was exchanged with

Results, p. 8, ll. 213 – 214: “Sample size, medial temporal lobe resection (MTLR) group: $n = 10$, control: $n = 20$; * = $p \leq 0.05$; ** = $p \leq 0.01$; *** = $p \leq 0.001$.”

12. Scatter: “average of all six distances”; relative to each other would be helpful for better understanding.

Thank you for your comment and we have made the following changes:

The following sentence

Results, p.6, ll. 164 – 166: “As a measure of precision, we computed the scatter of participants’ responses by calculating the average of all six distances between the four final locations per target location in the probe trials.”

was exchanged with

Results, p.6, ll. 181 – 183: “As a measure of precision, we computed the scatter of participants’ responses by calculating the relative distance of all six distances between the four final locations per target location in the probe trials.”

13. Fig.2: 1000 locations, following which distribution – spatial unit distribution

We are happy to clarify which distribution the random locations follow. The locations were uniformly distributed across the circular arena.

The following sentence

Results, p.7, ll. 179 – 180: “1,000 random locations were generated,...”

was changed to

Results, p.7, ll. 195 – 196: “1,000 random locations with uniform spatial distribution were generated.”

14. Pg 7: “reduction in average latency”; relative to the first one?

To clarify the meaning of the sentence we changed it from

Results, p. 8, ll. 205 - 208: “In learning trials, the improved temporal efficiency was reflected in a reduction in average latency to final location by 48.8% in patients (Mean \pm SEM: 46.7 \pm 17.4 vs. 23.9 \pm 4.9, supplementary table 1) and by 22.2% in controls (Mean \pm SEM: 19.8 \pm 2.3 vs. 15.4 \pm 1.1; setup: $F_{(1,28)} = 7.856$, $P = 0.009$, $\omega^2 = 0.19$).”

to

Results, p. 8, ll. 221- 224: “In learning trials, the improved temporal efficiency in the mobile setup compared to the stationary setup was reflected in a reduction in average latency to final location by 48.8% in patients (Mean \pm SEM: 46.7 \pm 17.4 vs. 23.9 \pm 4.9, supplementary table 1) and by 22.2% in controls (Mean \pm SEM: 19.8 \pm 2.3 vs. 15.4 \pm 1.1; setup: $F_{(1,27)} = 7.310$, $p = 0.012$, $\omega^2 = 0.18$).”

15. Discussion: Both, patients – insert a comma.

As suggested, we inserted a comma in the following sentence:

Discussion, p.15, ll. 393 - 395: “Both, patients with hippocampal lesions and healthy control participants showed overall improvements in memory-guided spatial navigation when multisensory input was available.”

#Reviewer 2

We thank the reviewer for the very helpful remarks on our manuscript and the suggestions made. We addressed all the raised issues and as a result made changes to the manuscript. Below is our response point by point.

1. Figure 1a: These pictures show participants wearing mobile EEG recording devices. But the manuscript did not mention anything about that except in the last section of the supplementary materials. I would suggest to clarify this in the main text with reasons for why EEG recordings were in this study but not reported in the paper.

The reviewer is correct, the experiment was conducted with EEG recordings. In the current study, we focused on the effect of hippocampal dysfunction and its impact on behavior with or without multisensory input to hypothesize the underlying brain mechanism.

In a second study we will test these newly generated hypotheses and perform a hypothesis-driven analysis of the recorded EEG data. Since data collection in patients with specific brain lesions is very time-consuming and our patients are all reintegrated into their professional life and live in different locations in Germany, we wanted to keep the effort for study participation for our patients very low. Therefore, we collected both behavioral and electrophysiological data simultaneously. Furthermore, the EEG data are very extensive and therefore beyond the scope of a single article. Instead, the EEG data obtained will be discussed in detail in a follow-up article.

We now describe the rationale in the methods section of the manuscript and added the following sentences to the section about the experimental setup:

Methods, p.20, "Experimental setup", ll.578 – 581: "Participants performed the task equipped with a fully mobile EEG system (supplementary material 1.7), the data from which will be reported in detail in a follow-up study with the focus on the electrophysiological dynamics during spatial navigation. EEG data will be analyzed to confirm or reject the hypotheses about brain dynamics raised in the current study."

2. Methods: Did the participants perform any disorientation between trials in the desktop condition? I did not see this information in the methods. Please add a little more description.

We thank the reviewer for this comment, as we did not indicate when a disorientation task was used or where it appeared. We now provide more detailed information about the disorientation task.

We changed the following sentences:

Results, p.4, 3rd paragraph, ll. 125 – 126: "After each learning or probe trial, a disorientation task followed that forced self-localization at the onset of each trial (see supplement 1.2, see methods)."

to

Results, p.4, 3rd paragraph, ll. 133 – 137: "After each learning or probe trial, a disorientation task followed that forced self-localization at the onset of each trial in both stationary and mobile setup."

Briefly, in this task, all spatial cues were blanked out. Participants first navigated to spheres that triggered a random sequence of three turns, then they were led by spheres to the starting point of the next trial and the spatial features of the virtual environment reappeared (see methods, see supplementary material 1.2).”

and

Methods, p. 19, “Behavioral testing”, ll. 558 – 559: “A disorientation task was inserted between all pairs of consecutive trials or after termination of a break between blocks... “

to

Methods, p.21, “Behavioral testing”, ll. 622 – 623: “A disorientation task was inserted between all pairs of consecutive trials or after termination of a break between blocks in both the stationary and mobile sessions.”

Furthermore, details of the disorientation task are described in the methods section, p. 21, “Behavioral testing”, 4th paragraph:

Methods, p.21, “Behavioral testing”, ll. 622 – 635: “A disorientation task was inserted between all pairs of consecutive trials or after termination of a break between blocks in both the stationary and mobile sessions. This was to prevent participants from using a simplistic strategy of immediately backtracking the learned trajectory from the previous trial. In the disorientation task, all visual features that could be used as a spatial cue were hidden, including the skybox. Participants were first asked to navigate to a waypoint – a blue sphere - at the center of the arena. Then a white sphere appeared in the viewing direction, which guided the participant to turn their body following a sequence of three rotations. The rotation sequence was randomized between right-left-right and left-right-left. After following the sequence of rotation, they were asked to walk straight to the starting location of the next trial indicated by a waypoint. Only then the sky and other spatially relevant features in the virtual environment were revealed again and the next trial started. The reasoning behind this manipulation was that the representation of the location of oneself formed in a trial should be “reset” at the beginning of the next one. As it is physically challenging to “teleport” participants in real world, we have rotated the virtual environment and masked the potential dissonance with the disorientation task. “

3. If I understand it correctly, the two conditions, desktop vs. immersive VR, were counterbalanced in order. If there is no order effect, please clarify this in the text.

We thank the reviewer for pointing this out. Indeed, the order was counterbalanced to control for a potential impact of time on task in general. Since the task was basically the same in both experimental setups, although the experimental environment was different, learning effects from the first experimental setup could potentially influence navigation behavior in the second experimental setup. We now included the order of sessions as a model covariate in the statistical model and changed the statistics in the manuscript accordingly.

The effect of session order on experimental variables was as follows:

Memory performance

Memory score: no effect, ($p = 0.828$, $\omega^2 = 0.0$)

Scatter of final locations: no effect, ($p = 0.673$, $\omega^2 = 0.0$)

Navigation efficiency

Latency to final location (learning trials): no effect ($p = 0.996$, $\omega^2 = 0.0$)

Latency to final location (probe trials): effect ($p = 0.008$, $\omega^2 = 0.20$)

Path error to final location (learning trials): no effect ($p = 0.992$, $\omega^2 = 0.0$)

Path error to final location (probe trials): effect $P < 0.001$ ($p < 0.001$, $\omega^2 = 0.41$)

Surface coverage (learning trials): no effect ($p = 0.183$, $\omega^2 = 0.03$)

Surface coverage (probe trials): effect $P = 0.002$ ($p = 0.002$, $\omega^2 = 0.28$)

Navigation strategy

Search accuracy (learning trials): no effect ($p = 0.285$, $\omega^2 = 0.00$)

Search accuracy (probe trials): no effect ($p = 0.810$, $\omega^2 = 0.02$)

Angular velocity (learning trials): no effect ($p = 0.557$, $\omega^2 = 0.00$)

Angular velocity (probe trials): no effect ($p = 0.517$, $\omega^2 = 0.00$)

Path replication (probe trials): no effect ($p = 0.689$, $\omega^2 = 0.00$)

In summary, the session order influenced navigation efficiency in the probe trials, in the sense that participants became more efficient at navigation as they gained experience with the task. We expected this and therefore counterbalanced the order of the setups. We added these results as an additional paragraph to the manuscript.

Results, p. 9, ll. 254 – 258: “In contrast to spatial memory performance and navigation strategies, we found an influence of the session order on navigation efficiency, at least for the performance in probe trials. The result indicates that with increasing experience with the task itself navigation efficiency increases (Probe trials: latency, $F_{(1,27)} = 8.096$, $p = 0.008$, $\omega^2 = 0.20$; path error, $F_{(1,27)} = 21.206$, $p < 0.001$, $\omega^2 = 0.41$; surface coverage, $F_{(1,27)} = 12.404$, $p = 0.002$, $\omega^2 = 0.28$; see supplementary table 4).”

Additionally, we added the following sentence:

Results, p. 4, ll. 119 – 121: “The session order of experimental setups was counterbalanced to account for potential learning effects from the first experimental setup that could influence navigation behavior in the second experimental setup.”

and we changed the following sentence:

Methods, pp. 23 – 24, “statistical analysis”, ll. 710 – 714: “Fixed effects were group (between-participants factor with two levels: MTLR and control) and setup (within-participants factor with two levels: stationary and mobile), and model covariates included participant sex, age, and years of education, and random effects included participant ID to account for interindividual differences.”

to

Methods, p.26, “Statistics and Reproducibility”, ll. 776 – 779: “Fixed effects were group (between-participants factor with two levels: MTLR and control) and setup (within-participants factor with two levels: stationary and mobile), and model covariates included session order, participant sex, age, and years of education, and random effects included participant ID to account for interindividual differences.”

And we added an additional table to the supplementary material (supplementary table 4):

Supplementary table 4. Table of results for influence of session order on experimental variables.		
	Learning trials	Probe trials
Spatial memory		
Memory score in percent	n.a.	$F_{(1,27)} = 0.048$, $p = 0.828$ $\omega^2 = 0.0$
Spatial precision		
Scatter as distance in virtual units	n.a.	$F_{(1,52)} = 0.181$, $p = 0.673$, $\omega^2 = 0.0$
Navigation efficiency		
Latency to final location in seconds	$F_{(1,27)} = 0.00$, $p = 0.996$, $\omega^2 = 0.0$	$F_{(1,27)} = 8.096$, $p = 0.008$, $\omega^2 = 0.20$
Path error to final location in percent	$F_{(1,52)} = 0.00$, $p = 0.992$, $\omega^2 = 0.0$	$F_{(1,27)} = 21.206$, $p < 0.001$, $\omega^2 = 0.41$
Surface coverage in percent	$F_{(1,27)} = 1.871$, $p = 0.183$, $\omega^2 = 0.03$	$F_{(1,27)} = 12.404$, $p = 0.002$, $\omega^2 = 0.28$
Navigation strategies		
Search accuracy/ avg. distance to final location in virtual units	$F_{(1,52)} = 1.166$, $p = 0.285$, $\omega^2 = 0.00$	$F_{(1,27)} = 1.462$, $p = 0.810$, $\omega^2 = 0.02$
Angular velocity/ idPhi (Integrated over the initial 5 seconds)	$F_{(1,27)} = 0.354$, $p = 0.557$, $\omega^2 = 0.00$	$F_{(1,52)} = 0.427$, $p = 0.517$, $\omega^2 = 0.00$
	n.a.	$F_{(1,52)} = 0.162$,

Path replication/
distance between paths
in virtual units

$p = 0.689,$
 $\omega^2 = 0.00$

4. I am not sure if this is a software issue but all “omega squared” signs for statistics in the pdf did not appear appropriately. I would also like to see how this effect size statistic was calculated.

We apologize for the inconvenience of not displaying the size of the omega square correctly in the PDF document. Since it is displayed correctly in the Word document, we assume that it was distorted during the conversion to the PDF document. We hope that this effect will not occur in the next version.

We chose the omega square instead of the eta square because we expect it to be less biased than the eta square for small sample sizes. The effect size was calculated with a predefined function of the freely available R package `effectsize` (https://easystats.github.io/effectsize/reference/eta_squared.html).

Here is an example:

```
library(effectsize)
omega_squared(model)
```

We have now included the description of the implemented effect size in the methods section:

Methods, p. 26, “Statistics and Reproducibility”, ll. 783 – 784: “The R package *effectsize* was used to calculate effect sizes as Omega squared (ω^2) (v.0.8.5).”

Furthermore, we now uploaded the R-functions used for statistical analysis to the online repository: <https://osf.io/u47mj/>

5. Page 22, Line 671: There is a typo “finalt” which should be “final”.

We thank the reviewer for pointing out this typo and we corrected it as suggested (methods, p. 24, l. 735).

6. The font size in the figures were a little too small.

We thank the reviewer for pointing out that the font size is too small. We have followed Nature's "Initial Submission" guideline. This recommends a font size of 5-7 <https://www.nature.com/nature/for-authors/initial-submission>, but we agree that a larger font size increases readability. Therefore, we increased the font size by 1 to 2 points.

a Stationary setup

Mobile setup

b Panorama view scene A

Panorama view scene B

c Learning trial

Disorientation

Probe trial

● Start location
● Target location

REVIEWERS' COMMENTS:

Reviewer #1 (Remarks to the Author):

All my comments were addressed properly and therefore I recommend publication of this submission.

I would like to take this opportunity to congratulate the authors for their well conducted research!

Reviewer #2 (Remarks to the Author):

Reviews for "Multisensory input modulates memory-guided spatial navigation" Communications Biology, COMMSBIO-23-1937A

I think the authors have done an excellent job of revising the manuscript. My previous questions have been well answered. I do not have further comments. I recommend this paper to be accepted.

Reply to the Reviewers

We would like to thank the editor and the reviewers for taking the time to read our revised manuscript thoroughly and are pleased that we were able to satisfy the reviewers.

Reviews for "Multisensory input modulates memory-guided spatial navigation"
Communications Biology, COMMSBIO-23-1937A

REVIEWERS' COMMENTS:

Reviewer #1 (Remarks to the Author):

All my comments were addressed properly and therefore I recommend publication of this submission. I would like to take this opportunity to congratulate the authors for their well conducted research!

Reviewer #2 (Remarks to the Author):

I think the authors have done an excellent job of revising the manuscript. My previous questions have been well answered. I do not have further comments. I recommend this paper to be accepted.